# Constant Degree Matrix-Driven Incomplete Multi-View Clustering via Connectivity-Structure and Embedding Tensor Learning

**Zhibin Gu**[1]*, **Zhenhao Zhong**[1]*, **Xi Zhang**[1], **Bing Li**[2,3]†

[1]College of Computer and Cyber Security, Hebei Normal University

[2]School of Computer Science, China University of Labor Relations

[3]State Key Laboratory of Multimodal Artificial Intelligence Systems,
Institute of Automation, Chinese Academy of Sciences
`guzhibin@hebtu.edu.cn`, `bli@nlpr.ia.ac.cn`

## Abstract

Tensor-based incomplete multi-view clustering has attracted significant research attention due to its capability to exploit high-order correlations across different views for revealing underlying cluster structures from partially observed multi-view data. However, most existing approaches construct tensors from adjacency matrices, which necessitate post-processing operations (e.g., singular value decomposition, SVD) and thereby introduce additional computational overhead and potential errors. Some approaches instead employ latent embedding tensors to avoid post-processing, but they often fail to capture the geometric structure of the underlying graph. To address these limitations, we propose **C**onst**A**nt degree **M**trix-driv**E**n incomp**L**ete multi-view clustering via connectivity-structure and embedding tensor learning (**CAMEL**). Specifically, CAMEL jointly learns view-specific latent embeddings under structured constraints and organizes them into a tensor with an $\ell_\delta$ low-rank constraint, thereby enabling coordinated optimization of graph connectivity and high-order correlations. To further mitigate the $\mathcal{O}(n^2)$ or ever higher complexity associated with conventional connectivity constraints, CAMEL approximates the variable Laplacian degree matrix with a constant-degree matrix, reducing the computational cost of degree matrix construction to $\mathcal{O}(1)$. Clustering assignments are subsequently derived via $k$-means on the concatenated embeddings, eliminating the need for post-processing operations on adjacency matrices such as SVD. Extensive experiments on nine benchmark datasets demonstrate the superior effectiveness and efficiency of CAMEL.

## 1 Introduction

Multi-view data are widely encountered in real-world tasks, where each sample can typically be represented from multiple perspectives or modalities. For example, an image can be represented from multiple perspectives, where features derived from texture, color, and local binary patterns are treated as separate views Wen et al. (2022); Gu et al. (2024). Multi-view clustering (MVC) aims to integrate information from different views to accurately uncover the intrinsic structure of the data Tang et al. (2020); Kang et al. (2020); Jiang et al. (2022). In practice, however, some views may be missing; for example, certain events may be reported by some sources but absent in others, resulting in incomplete multi-view data. shihe To effectively handle such incomplete multi-view data, numerous incomplete multi-view clustering (IMVC) methods have been proposed, which can be broadly categorized into matrix-based Wen et al. (2019); Liu et al. (2018); Li et al. (2021) and tensor-based approaches Li et al. (2024); Yu et al. (2024); Li et al. (2023b). Among them, tensor-based methods have attracted widespread attention due to their ability to effectively model high-order correlations across multiple views. For example, Gu & Feng (2024) imposes low-rank regularization on the

---

*These authors contributed equally.
†Corresponding author.

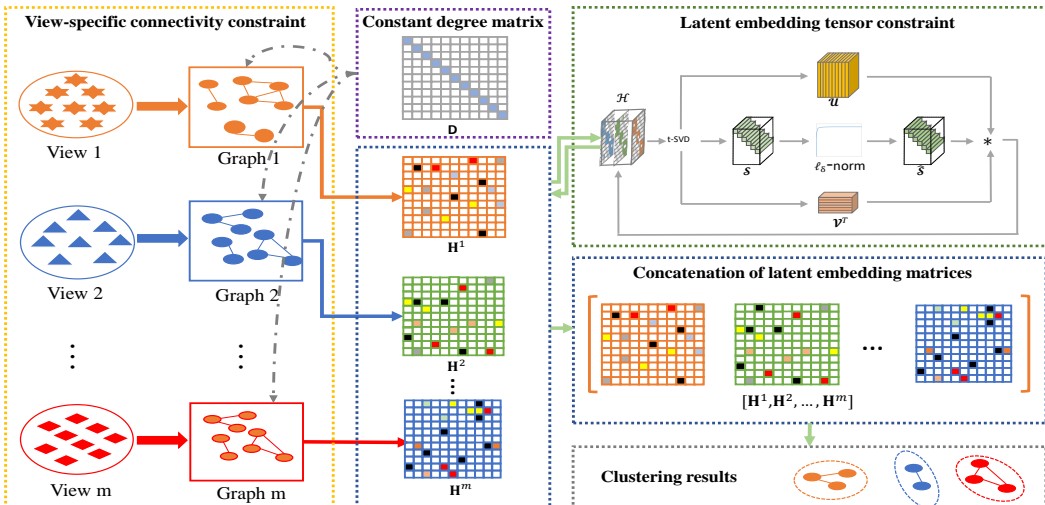

Figure 1: Overall framework of the proposed CAMEL model, which integrates connectivity and low-rank constraints on the latent embedding tensor, with a constant degree matrix employed in the Laplacian to reduce computational complexity.

tensor constructed from coefficient and projection matrices, thereby reducing computational complexity and enhancing robustness. In addition, Zhang et al. (2023) employs a decomposition strategy that separates the coefficient tensor into low-rank and sparse noise components for independent optimization. However, these approaches require post-processing[1] (e.g., SVD) of the coefficient matrices to extract the latent embedding representation for subsequent $k$-means clustering. The decoupling of graph learning and latent embedding learning may lead to suboptimal performance due to potential error propagation between the two stages. To eliminate the need for post-processing, recent tensor-based methods such as Xu et al. (2025); Liu et al. (2025) construct tensors from latent embedding matrices, allowing $k$-means to be directly applied on them to obtain clustering results. Despite this improvement, a major limitation remains: the absence of connectivity constraints prevents the explicit preservation of the global geometric structure within each view. While imposing connectivity constraints on the adjacency matrix is a natural solution, this typically requires constructing the Laplacian matrix, which involves computing the degree matrix from the similarity matrix, resulting in a time complexity of $\mathcal{O}(n^2)$ or higher and significantly limiting computational efficiency. Moreover, since the latent embedding is represented as a single consensus matrix across views, existing connectivity-based methods cannot naturally accommodate tensor-based constraints, making it difficult to simultaneously capture both the low-rank properties of the latent embedding and the potential high-order correlations across multiple views.

To address these limitations, we propose an effective and novel model, CAMEL, whose overall framework is detailed in Figure 1. The core innovation of CAMEL lies in the introduction of a constant-degree matrix instead of a variable-degree matrix in the Laplacian construction. This design preserves the essential geometric structure of the graph while reducing the computational cost of connectivity-constrained embedding learning from $\mathcal{O}(n^2)$ to $\mathcal{O}(1)$. Moreover, CAMEL integrates the constant-degree-driven connectivity constraint with a low-rank constraint on the latent embedding tensor, enabling the model to learn the geometric structure of each view and construct a tensor that simultaneously captures low-rank properties and high-order correlations across views. The resulting latent embeddings can be directly used for clustering without post-processing, thereby avoiding potential errors introduced by additional steps. Compared to existing methods, our key contributions can be summarized as follows:

---

[1] The term "post-processing-free" refers to the fact that the features obtained after optimization can be directly used for k-means clustering to obtain results, without requiring additional operations on these features (e.g., SVD on the anchor subspace representation matrix or performing spectral clustering on the similarity matrix) to generate new features for clustering.

- We approximate the variable Laplacian degree matrix in the connectivity constraint with a constant-degree matrix, reducing computational complexity by an order of magnitude while preserving competitive accuracy and scalability to large datasets.

- We integrate the constant-degree-driven connectivity constraint with a low-rank constraint on the latent embedding tensor, enabling the model to learn the geometric structure of each view and construct a tensor that simultaneously captures low-rank properties and high-order correlations across views.

- We develop an efficient algorithm for jointly solving the latent embedding matrices within an ADMM-based optimization framework and provide theoretical guarantees for its convergence. Extensive experiments validate the effectiveness and efficiency of CAMEL.

## 2 RELATED WORK

Tensors are well-suited for capturing high-order correlations, which makes them particularly effective for imputing missing data and has led to their widespread application in incomplete multi-view clustering (IMVC). Existing tensor-based IMVC methods can generally be categorized into two groups: post-processing-based approaches and post-processing-free approaches.

For the post-processing-based methods, latent embeddings are typically obtained by applying SVD to the anchor subspace representation matrix or by performing spectral clustering on the similarity matrix, after which $k$-means is conducted on the latent embeddings to derive the final clustering results Wen et al. (2021); Zhang et al. (2015); Zhao et al. (2023); Long et al. (2025; 2024); Lv et al. (2023); Zhang et al. (2023); Li et al. (2022b). For instance, Long et al. (2024) obtains per-view embedding features by projecting the anchor graph onto the designated matrix, and then applies a tensor low-frequency approximation to the tensor constructed from these embedding feature matrices. Wen et al. (2021) enhances representation-based graph learning by simultaneously imposing a consensus graph constraint and a tensor low-rank regularization, thereby capturing both within-view and cross-view dependencies. Moreover, Lv et al. (2023) maps the original data into a low-dimensional feature subspace to achieve more compact representations, which facilitates subsequent graph construction.

As for the post-processing-free methods, they directly obtain the latent embedding matrix after optimization, and clustering results are then produced by applying $k$-means on this matrix Li et al. (2023a); Xu et al. (2025); Liu et al. (2025); Wang et al. (2025); Chen et al. (2022); Wu et al. (2024). For example, Li et al. (2023a) introduces an orthogonal non-negative tensor factorization framework that extends traditional NMF to third-order tensors, enabling direct handling of multi-view data while preserving and exploiting the inherent spatial structures across views. In addition, Xu et al. (2025) learns anchor–sample similarity graphs in an adaptive manner, enforces structural refinement in a latent subspace, and employs low-rank modeling on the latent sample embeddings. In Liu et al. (2025), the method eliminates the tensor rotation trick to prevent its interference with unsupervised clustering, and further leverages pairwise sample similarities via a linear kernel function to ensure scalability to large datasets.

## 3 PROPOSED METHOD

### 3.1 ANCHOR SUBSPACE-BASED TENSOR REPRESENTATION LEARNING

Let the incomplete multi-view dataset be denoted as $\{\mathbf{X}^v\}_{v=1}^m$, where $\mathbf{X}^v \in \mathbb{R}^{d_v \times n}$, with $d_v$ representing the feature dimension of the $v$-th view, $n$ is the total number of samples, and $m$ the total number of views. In anchor-based tensor incomplete multi-view clustering, the anchor graphs learned from different views are combined into a three-dimensional tensor, and a post-processing step is performed on these anchor graphs prior to final clustering to obtain the cluster assignments. The general framework is summarized as follows:

$$
\min_{\{\mathbf{E}^v, \mathbf{Z}^v, \mathbf{A}^v\}_{v=1}^m} \mathcal{R}(\boldsymbol{\mathcal{Z}}) + \lambda_1 \mathcal{P}(\mathbf{E}^v)
$$
$$
\text{s.t. } \forall v, \ \mathbf{X}^v = \mathbf{A}^v \mathbf{Z}^v + \mathbf{E}^v, \mathbf{Z}^v \geq 0, (\mathbf{A}^v)^\top \mathbf{A}^v = \mathbf{I}, \boldsymbol{\mathcal{Z}} = \Phi(\mathbf{Z}^1, \mathbf{Z}^2, \dots, \mathbf{Z}^m)
$$
(1)

where $\mathbf{A}^v \in \mathbb{R}^{d_v \times t}$, $\mathbf{Z}^v \in \mathbb{R}^{t \times n}$, and $\mathbf{E}^v \in \mathbb{R}^{d_v \times n}$ denote the anchor matrix, coefficient matrix, and reconstruction error matrix of the $v$-th view, respectively, with $t$ representing the number of anchors.

To ensure discriminativeness among anchors, an orthogonality constraint $(\mathbf{A}^v)^\top \mathbf{A}^v = \mathbf{I}$ is imposed to enhance anchor diversity. Physically, $\mathbf{Z}^v$ is constrained to be nonnegative, i.e., $\mathbf{Z}^v \geq 0$. The tensor $\mathcal{Z}$ is constructed by concatenating and rotating $\{\mathbf{Z}^v\}_{v=1}^m$ via the operation $\Phi$. The regularization term $\mathcal{R}(\cdot)$ models the tensor rank and promotes low-rank properties during optimization, while $\mathcal{P}(\cdot)$ regularizes the reconstruction error to enforce $\mathbf{X}^v \approx \mathbf{A}^v \mathbf{Z}^v$.

## 3.2 JOINT LEARNING OF CONNECTIVITY-CONSTRAINED EMBEDDING TENSOR

Although the incorporation of tensors is capable of capturing the high-order correlations across multiple views, it is still insufficient to explicitly characterize the underlying graph structure. First, since $\{\mathbf{Z}^v\}_{v=1}^m$ cannot be directly employed for clustering, a post-processing step (e.g., SVD) is required to derive the cluster labels. Such reliance on post-processing not only increases the risk of obtaining suboptimal solutions but also introduces additional computational overhead. Second, Eq. (1) fails to explicitly capture the underlying graph structure, resulting in the loss of neighborhood information and reduced discriminability of the learned representations.

To address these limitations, we introduce two key enhancements. First, the connectivity constraint is reformulated in a view-specific manner and integrated into the subspace structure. Second, motivated by recent advances Xu et al. (2025); Wang et al. (2025); Liu et al. (2025); Li et al. (2023a); Feng et al. (2024), we impose a low-rank constraint directly on the tensor constructed from latent embeddings, rather than on that derived from subspace representations. This design enables $k$-means clustering to be directly performed on the latent embedding matrices, thereby avoiding the costly SVD computation required for subspace representations. Moreover, the rank function $\mathcal{R}(\cdot)$ is approximated by the $\ell_\delta$-norm (i.e., $f(x) = \frac{(1+\delta)x}{\delta+x}$) Kang et al. (2015), which offers a smoother and tighter surrogate than the conventional tensor nuclear norm Lu et al. (2016); Zhou et al. (2019); Lu et al. (2019) by penalizing singular values in a more flexible manner. In addition, the squared Frobenius norm $\|\cdot\|_F^2$ is employed to model the noise matrix. The overall model is thus formulated as follows:

$$\min_{\{\mathbf{A}^v, \mathbf{E}^v, \mathbf{Z}^v, \mathbf{H}^v\}_{v=1}^m} \|\mathcal{H}\|_{\ell_\delta} + \lambda_1 \sum_{v=1}^m \|\mathbf{E}^v\|_F^2 + \lambda_2 \sum_{v=1}^m \mathrm{Tr}((\mathbf{H}^v)^\top \mathbf{L}^v \mathbf{H}^v) \qquad (2)$$
$$s.t.\ \mathbf{X}^v = \mathbf{A}^v \mathbf{Z}^v + \mathbf{E}^v, \mathbf{Z}^v \geq 0, (\mathbf{A}^v)^\top \mathbf{A}^v = \mathbf{I}, (\mathbf{H}^v)^\top \mathbf{H}^v = \mathbf{I}, \mathcal{H} = \Phi(\mathbf{H}^1, \mathbf{H}^2, \cdots, \mathbf{H}^m)$$

where $\mathbf{L}^v$ is the normalized Laplacian matrix, i.e., $\mathbf{L}^v = \mathbf{I} - (\mathbf{D}^v)^{-\frac{1}{2}} \mathbf{S}^v (\mathbf{D}^v)^{-\frac{1}{2}}$. $\mathbf{S}^v \in \mathbb{R}^{n \times n}$ is the affinity matrix, obtained by $\mathbf{S}^v = (\mathbf{Z}^v)^\top \mathbf{Z}^v$, and $\mathbf{D}^v \in \mathbb{R}^{n \times n}$ is the degree matrix computed by summing the columns of $\mathbf{S}^v$. The two hyperparameters $\lambda_1$ and $\lambda_2$ serve solely for trade-off purposes. For a third-order tensor $\mathcal{H} \in \mathbb{R}^{n_1 \times n_2 \times n_3}$, its $\ell_\delta$-norm is mathematically expressed as follows:

$$\|\mathcal{H}\|_{\ell_\delta} = \frac{1}{n_3} \sum_{k=1}^{n_3} \|\mathcal{H}_f^k\|_{\ell_\delta} = \frac{1}{n_3} \sum_{k=1}^{n_3} \sum_{i=1}^h \frac{(1+\delta)\boldsymbol{\mathcal{S}}_f^k(i,i)}{\delta + \boldsymbol{\mathcal{S}}_f^k(i,i)}$$

In this formulation, $h$ is defined as the minimum of $n_1$ and $n_2$, while the positive scalar $\delta$ serves as a parameter of the $\ell_\delta$ rank function. The matrix $\boldsymbol{\mathcal{S}}_f^k$ corresponds to the $k$-th frontal slice of the tensor $\boldsymbol{\mathcal{S}}_f$, derived from the tensor singular value decomposition (t-SVD) applied to the $k$-th frontal slice of $\mathcal{H}$, following the model $\mathcal{H}_f^k = \boldsymbol{\mathcal{U}}_f^k \boldsymbol{\mathcal{S}}_f^k (\boldsymbol{\mathcal{V}}_f^k)^\top$.

## 3.3 CONSTANT DEGREE MATRIX-DRIVEN CONNECTIVITY-TENSOR JOINT LEARNING

Although Eq.2 seamlessly unifies latent representation tensor learning with connectivity constraints, the construction of the Laplacian matrix remains computationally expensive with a complexity of $\mathcal{O}(n^2)$. In particular, evaluating $\mathbf{S}^v = (\mathbf{Z}^v)^\top \mathbf{Z}^v$ incurs $\mathcal{O}(tn^2)$ operations. While reordering the computational steps during optimization can reduce this cost to linear in $\mathcal{O}(n)$, the computation of the degree matrix $\mathbf{D}^v$ still requires column-wise summation of $\mathbf{S}^v \in \mathbb{R}^{n \times n}$, which scales as $\mathcal{O}(n^2)$. This quadratic overhead inevitably becomes a bottleneck when dealing with large-scale datasets. To address this issue, we replace the conventional variable degree matrix with a constant form, $\mathbf{D} = \beta \mathbf{I}$, where $\beta > 0$ controls the degree magnitude. Consequently, the Laplacian matrix is simplified to $\mathbf{L}^v = \mathbf{I} - \frac{1}{\beta}(\mathbf{Z}^v)^\top \mathbf{Z}^v$. Based on this, the overall CAMEL model can be expressed as follows:

$$\min_{\{\mathbf{A}^v,\mathbf{E}^v,\mathbf{Z}^v,\mathbf{H}^v\}_{v=1}^m} \|\mathcal{H}\|_{\ell_\delta} + \lambda_1 \sum_{v=1}^m \|\mathbf{E}^v\|_F^2 + \lambda_2 \sum_{v=1}^m \mathrm{Tr}[(\mathbf{H}^v)^\top (\mathbf{I} - \frac{1}{\beta}(\mathbf{Z}^v)^\top \mathbf{Z}^v)\mathbf{H}^v] \tag{3}$$

$$s.t. \ \mathbf{X}^v = \mathbf{A}^v \mathbf{Z}^v + \mathbf{E}^v, \mathbf{Z}^v \geq 0, (\mathbf{A}^v)^\top \mathbf{A}^v = \mathbf{I}, (\mathbf{H}^v)^\top \mathbf{H}^v = \mathbf{I}, \mathcal{H} = \Phi(\mathbf{H}^1, \mathbf{H}^2, \cdots, \mathbf{H}^m)$$

### 3.4 FURTHER DISCUSSION ON THE CAMEL MODEL

The CAMEL model introduces a key innovation by efficiently integrating connectivity constraints with the low-rank learning of latent embedding tensors through a constant degree matrix approximation. This design not only significantly reduces computational complexity but is also justified from the following three perspectives. First, since $\mathbf{Z}^v$ inherently contains noise, the degree matrix computed from $(\mathbf{Z}^v)^\top \mathbf{Z}^v$ is itself inaccurate. Therefore, employing a suitably approximated degree matrix does not necessarily impair clustering performance more than using $(\mathbf{Z}^v)^\top \mathbf{Z}^v$. Second, the model is able to obtain relatively clean $\mathbf{H}^v$, which is the representation actually used for clustering, rather than the noisier $\mathbf{Z}^v$. As connected component partitioning is not a highly fine-grained operation, once the adjacency graph captures the coarse structure, the resulting $\mathbf{H}^v$ is sufficiently reliable. Last but not least, the tensor constraint on $\mathcal{H}$ can further filter out noise introduced by the approximated degree matrix, reconstruction errors and missing data. By constructing the degree matrix through a constant approximation, the computational complexity of computing $\mathbf{D}^v$ is reduced from $\mathcal{O}(n^2)$ to $\mathcal{O}(1)$, lowering the overall cost. This provides important insights into designing connectivity and tensor joint constraints based on latent embedding matrices.

## 4 THEORETICAL ANALYSIS

### 4.1 OPTIMAZITION

We employ the ADMM Lin et al. (2011) to optimize Eq.(2). Accordingly, the corresponding augmented Lagrangian function can be formulated as follows:

$$\mathcal{L}(\{\mathbf{A}^v\}_{n=1}^m, \{\mathbf{E}^v\}_{v=1}^m, \{\mathbf{Z}^v\}_{v=1}^m, \{\mathbf{H}^v\}_{v=1}^m, \{\mathbf{Y}^v\}_{v=1}^m, \{\mathbf{B}^v\}_{v=1}^m, \mathcal{W}, \mathcal{G})$$

$$= \|\mathcal{G}\|_{\ell_\delta} + \lambda_1 \sum_{v=1}^m \|\mathbf{E}^v\|_F^2 + \lambda_2 \sum_{v=1}^m \mathrm{Tr}[(\mathbf{H}^v)^\top (\mathbf{I} - \frac{1}{\beta}(\mathbf{Z}^v)^\top \mathbf{Z}^v)\mathbf{H}^v] + \langle \mathcal{W}, \mathcal{H} - \mathcal{G} \rangle \tag{4}$$

$$+ \frac{\zeta}{2}\|\mathcal{H} - \mathcal{G}\|_F^2 + \sum_{v=1}^m \left( \langle \mathbf{Y}^v, \mathbf{X}^v - \mathbf{A}^v \mathbf{Z}^v - \mathbf{E}^v \rangle + \frac{\mu}{2}\|\mathbf{X}^v - \mathbf{A}^v \mathbf{Z}^v - \mathbf{E}^v\|_F^2 \right)$$

where $\mu$ and $\zeta$ are positive parameters, $\mathcal{G}$ denotes an auxiliary tensor, and $\mathcal{W}$ together with $\mathbf{Y}^v$ serve as Lagrange multipliers. The complete optimization procedure is presented in the appendix.

### 4.2 CONVERGENCE ANALYSIS

Convergence of the optimization algorithm is theoretically ensured by Theorem 1, with the appendix offering a more comprehensive discussion of the theoretical foundations and implementation details.

**Theorem 1** *The iterative sequence $\{\mathcal{J}_p = \mathbf{A}_p^v, \mathbf{Z}_p^v, \mathbf{E}_p^v, \mathbf{H}_p^v, \mathbf{Y}_p^v, \mathcal{G}_p, \mathcal{W}_p\}_{p=1}^\infty$ produced by our optimization scheme exhibits the following convergence properties:*

- *The sequence $\{\mathcal{J}_p\}_{p=1}^\infty$ remains bounded;*

- *Every accumulation point of $\{\mathcal{J}_p\}_{p=1}^\infty$ is guaranteed to be a stationary point satisfying the KKT conditions.*

### 4.3 COMPLEXITY ANALYSIS

The overall computational burden of our model is primarily determined by the optimization of five sets of variables. These variables include $\mathbf{A}^v$, $\mathbf{Z}^v$, $\mathbf{E}^v$, $\mathbf{H}^v$, and $\mathcal{G}$, with their respective complexities

given by $\mathcal{O}(d_v tn + d_v t)$, $\mathcal{O}(d_v n + d_v tn + tn)$, $\mathcal{O}(d_v tn + d_v n)$, $\mathcal{O}(tcn + c^2 n)$, and $\mathcal{O}(mtn \log(mn) + m^2 tn)$. Aggregating these costs yields an overall computational complexity of $\mathcal{O}(mtn \log(n) + d_{max} tn)$, with $d_{max}$ indicating the largest feature dimension across the $m$ views. It indicates that our approach achieves a substantially efficient complexity.

## 5 EXPERIMENTS

### 5.1 EXPERIMENTAL SETTINGS

All experimentals are implemented in MATLAB R2023b and executed on a standard computing platform with an i7-12650H CPU and 16 GB RAM. The core findings are presented in this section, while additional results and detailed analyses can be found in the Appendix.

Table 1: The datasets used in the experiments.

| Dataset | Samples | Views |
|---|---|---|
| Yale3 | 165 | 3 |
| MSRCV1 | 210 | 5 |
| Still-DB | 467 | 3 |
| EYaleB10 | 640 | 3 |
| COIL20MV | 1440 | 4 |
| Mfeat | 2000 | 6 |
| Scene | 2688 | 4 |
| Scene15 | 4485 | 3 |
| Noisy MNIST | 10000 | 2 |
| CIFAR10 | 50000 | 3 |

**Datasets:** We conduct experiments on nine publicly available multi-view datasets. These datasets cover diverse domains including face images(Yale3(Belhumeur et al. (1997)), EYaleB10(Liu et al. (2010))), an action image dataset(Still-DB(Ikizler et al. (2008))), scene images(Scene(Fang et al. (2024)), Scene15(Fei-Fei & Perona (2005))), object recognition(MSRCV1(Xia et al. (2022a)), COIL20MV, CIFAR10(Liu et al. (2022))), as well as handwritten digits(Mfeat(Nie et al. (2018)), Noisy MNIST). The number of samples and views for each dataset are summarized in Table 1.

**Comparison methods:** Six baseline methods are selected for comparison, namely BSV (2001)(Ng et al. (2001); Wen et al. (2018; 2022)), Concat(Wen et al. (2018; 2022)), PVC (2014)(Li et al. (2014)), IMVC-CBG (2022)(Wang et al. (2022)), PSIMVC-PG (2024)(Li et al. (2022a)), and SCSL (2024)(Liu et al. (2024)). For those methods that require hyperparameter tuning, the parameters are adjusted within the recommended ranges, whereas for our model, the search is conducted over $\lambda_1 \in [10^{-8}, 10^{-3}]$, $\lambda_2 \in [10^{-4}, 10^{-2}]$, $\beta \in [10^1, 10^4]$, and $\delta \in [10^{-3}, 10^0]$.

**Evaluation metrics:** To guarantee the robustness of the evaluation, each algorithm is executed five times. The clustering results are then measured by three criteria: ACC, NMI, and PUR.

**Construction of incomplete multi-view data:** To simulate incomplete multi-view data, we introduce missing entries by randomly discarding a proportion of samples in each view, where the missing ratio $r$ is selected from the set $\{0.1, 0.3, 0.5\}$. In order to avoid the situation in which a sample disappears from all views, one view is randomly designated to retain the corresponding data whenever such a case occurs. This procedure guarantees that every instance within the constructed incomplete multi-view dataset possesses at least one observed representation. All datasets are first processed into incomplete-view versions, and these incomplete-view datasets remain unchanged in subsequent experiments. For example, the original Yale3 dataset is first processed into an incomplete-view dataset Yale3_i, and all subsequent experiments are conducted on this fixed Yale3_i without further modifications. Unless otherwise stated (specifically in Section 5.2), the experiments are carried out with a default setting of $r = 0.5$.

### 5.2 CLUSTERING RESULTS

**Performance comparison:** Table 2 reports the comparative outcomes of different clustering methods across all datasets, where the best-performing result in each case is emphasized in **bold**, and the runner-up result is annotated with underline. The notation "OM" denotes an out-of-memory error. From the overall analysis of these clustering outcomes, two key insights can be drawn as follows:

(1) Our method achieves consistently superior performance across diverse datasets, including face recognition, object categorization, scene classification, handwritten digits, action recognition, and large-scale collections. In nearly all cases and under different missing rates, the proposed approach substantially outperforms baseline methods in terms of ACC, NMI, and PUR. For example, on the MSRCV1 dataset with a missing rate of 0.1, CAMEL exceeds the second-best method by 23.05%, 29.03%, and 23.05% in ACC, NMI, and PUR, respectively. Furthermore, on datasets such as

Table 2: Comparison of clustering results across varying levels of missing data.

| Data | Methods | Missing rate 0.1 | | | 0.3 | | | 0.5 | | |
|---|---|---|---|---|---|---|---|---|---|---|
| | | ACC | NMI | PUR | ACC | NMI | PUR | ACC | NMI | PUR |
| Yale3 | BSV | 35.64±6.99 | 41.06±6.93 | 36.73±6.74 | 33.82±6.17 | 35.53±6.85 | 35.76±5.88 | 26.06±2.94 | 25.59±3.36 | 28.48±2.74 |
| | Concat | 31.76±11.65 | 31.99±12.17 | 34.30±11.25 | 28.12±4.57 | 28.98±6.48 | 30.42±5.01 | 22.55±5.82 | 20.43±6.90 | 24.48±5.40 |
| | PVC | 50.30±2.54 | 53.58±2.06 | 51.27±1.80 | 39.74±2.22 | 43.10±1.43 | 40.90±2.19 | 40.44±1.86 | 42.82±1.55 | 42.22±2.16 |
| | IMVC-CBG | 44.48±0.33 | 45.75±0.00 | 45.09±0.33 | 37.58±0.00 | 38.55±0.00 | 38.79±0.00 | 22.91±0.27 | 19.42±0.29 | 23.52±0.27 |
| | PSIMVC-PG | 52.73±0.00 | 56.24±0.00 | 53.94±0.00 | 31.39±0.66 | 35.22±1.06 | 32.97±0.81 | 19.39±0.00 | 18.38±0.00 | 21.82±0.00 |
| | SCSL | 63.03±0.00 | 64.79±0.00 | 63.03±0.00 | 55.76±0.00 | 58.61±0.00 | 55.76±0.00 | 32.12±0.00 | 35.75±0.00 | 33.94±0.00 |
| | **CAMEL** | **74.79 ±4.90** | **79.25 ±3.19** | **75.64 ±3.41** | **72.61±5.02** | **76.21 ±4.15** | **73.21 ±5.13** | **61.82±2.27** | **65.17 ±2.53** | **63.03 ±2.54** |
| MSRCV1 | BSV | 60.00±9.61 | 51.44±5.29 | 62.86±6.01 | 40.10±4.71 | 28.93±3.48 | 41.81±3.99 | 31.43±3.10 | 18.12±2.97 | 32.19±3.06 |
| | Concat | 73.33±10.43 | 66.58±7.61 | 74.29±9.39 | 54.29±7.66 | 47.78±5.41 | 56.86±6.41 | 44.10±2.58 | 33.60±4.19 | 44.86±2.78 |
| | PVC | 62.78±5.77 | 49.46±3.47 | 64.98±3.36 | 71.24±5.34 | 59.36±3.80 | 71.96±4.53 | 52.25±9.29 | 46.75±8.86 | 56.75±8.47 |
| | IMVC-CBG | 50.19±0.43 | 41.06±0.75 | 52.10±0.64 | 36.67±0.00 | 24.83±0.00 | 37.62±0.00 | 19.05±0.00 | 4.80±0.00 | 19.05±0.00 |
| | PSIMVC-PG | 46.67±0.00 | 36.14±0.00 | 47.62±0.00 | 28.76±0.26 | 17.34±0.26 | 29.71±0.26 | 18.29±0.43 | 5.50±0.44 | 19.33±0.43 |
| | SCSL | 74.76±0.00 | 64.82±0.00 | 74.76±0.00 | 61.90±0.00 | 56.05±0.00 | 66.19±0.00 | 67.14±0.00 | 61.87±0.00 | 70.48±0.00 |
| | **CAMEL** | **97.81 ±0.43** | **95.61 ±0.58** | **97.81 ±0.43** | **97.14 ±0.95** | **94.05 ±1.39** | **97.14 ±0.95** | **97.90 ±0.26** | **96.03 ±0.59** | **97.90 ±0.26** |
| Still-DB | BSV | 29.29±1.19 | 10.91±0.53 | 32.98±0.82 | 29.68±1.25 | 8.93±0.71 | 31.78±1.45 | 25.65±0.63 | 7.72±0.78 | 27.11±0.42 |
| | Concat | 31.82±1.43 | 11.25±0.66 | 33.83±1.03 | 29.08±1.41 | 10.13±1.14 | 32.81±1.57 | 28.27±1.02 | 9.09±1.08 | 29.94±1.19 |
| | PVC | 21.86±0.00 | 0.00±0.00 | 21.86±0.00 | 21.74±0.00 | 0.00±0.00 | 21.74±0.00 | 22.52±0.00 | 0.00±0.00 | 22.52±0.00 |
| | PSIMVC-PG | 29.34±0.00 | 9.29±0.00 | 32.55±0.00 | 28.27±0.00 | 8.08±0.00 | 30.62±0.00 | 28.69±0.00 | 8.99±0.00 | 31.26±0.00 |
| | IMVC-CBG | 32.29±0.01 | 10.97±0.00 | 33.79±0.00 | 29.34±0.00 | 8.14±0.00 | 31.05±0.00 | 29.94±0.18 | 8.94±0.10 | 32.08±0.18 |
| | SCSL | 25.27±0.00 | 3.49±0.00 | 26.34±0.00 | 22.48±0.00 | 2.62±0.00 | 23.77±0.00 | 26.55±0.00 | 5.88±0.00 | 28.27±0.00 |
| | **CAMEL** | **59.36 ± 1.88** | **45.34 ±1.41** | **61.88±1.60** | **67.45 ±0.30** | **53.47 ±0.39** | **67.45 ±0.30** | **74.78 ±0.46** | **66.61 ±0.40** | **74.78 ±0.46** |
| EYaleB10 | BSV | 25.44±1.00 | 23.32±2.42 | 27.81±0.98 | 18.09±0.92 | 7.62±1.20 | 18.81±1.16 | 21.53±1.37 | 14.28±1.77 | 22.81±1.38 |
| | Concat | 17.53±2.24 | 6.83±3.47 | 19.03±2.92 | 18.59±1.57 | 8.13±2.56 | 19.47±1.94 | 17.75±2.05 | 5.40±1.98 | 17.75±1.73 |
| | PVC | 36.25±5.47 | 35.07±7.80 | 37.76±4.71 | 31.09±1.21 | 27.32±2.40 | 32.75±1.56 | 30.40±2.44 | 25.56±3.89 | 31.73±2.59 |
| | IMVC-CBG | 33.94±0.13 | 28.47±0.11 | 34.72±0.13 | 27.19±0.11 | 18.61±0.00 | 28.13±0.11 | 17.25±0.14 | 7.86±0.12 | 18.66±0.14 |
| | PSIMVC-PG | 30.09±0.00 | 23.88±0.12 | 31.34±0.00 | 24.06±0.00 | 15.20±0.00 | 24.69±0.00 | 17.03±0.00 | 8.78±0.00 | 19.53±0.00 |
| | SCSL | 12.81±0.00 | 3.31±0.00 | 13.13±0.00 | 12.19±0.00 | 2.72±0.00 | 12.50±0.00 | 20.00±0.00 | 8.42±0.00 | 20.78±0.00 |
| | **CAMEL** | **51.97 ±1.78** | **51.09 ±0.98** | **52.44 ±1.58** | **48.59 ±4.73** | **47.11 ±4.73** | **48.94 ±4.49** | **55.72 ±5.42** | **51.21 ±5.48** | **55.78 ±5.35** |
| COIL20MV | BSV | 52.81±4.93 | 65.04±2.26 | 56.89±4.05 | 42.06±3.63 | 49.45±1.51 | 44.82±2.85 | 31.74±1.95 | 37.18±1.58 | 33.93±1.86 |
| | Concat | 58.68±7.56 | 73.38±3.21 | 63.65±6.47 | 47.47±1.97 | 58.62±1.44 | 51.32±1.72 | 37.67±3.54 | 47.53±3.99 | 41.08±3.09 |
| | PVC | 5.05±0.00 | 0.00 ± 0.00 | 5.05 ±0.00 | 5.29 ± 0.00 | 0.00 ± 0.00 | 5.29 ±0.00 | 5.65 ± 0.00 | 0.00 ± 0.00 | 5.65 ± 0.00 |
| | IMVC-CBG | 56.51±1.30 | 67.58±0.59 | 59.63±0.85 | 50.90±0.47 | 58.12±0.28 | 54.61±0.26 | 41.00±1.40 | 50.45±1.00 | 44.18±1.29 |
| | PSIMVC-PG | 56.79±1.85 | 67.84±0.53 | 60.08±1.25 | 50.69±0.16 | 58.01±0.22 | 54.42±0.35 | 32.81±0.57 | 39.39±0.42 | 36.44±0.50 |
| | SCSL | 26.81±0.00 | 28.51±0.00 | 30.00±0.00 | 52.71±0.00 | 62.34±0.00 | 55.26±0.00 | 40.69±0.00 | 51.00±0.00 | 46.32±0.00 |
| | **CAMEL** | **68.38 ±2.27** | **79.17 ±0.98** | **71.22 ±1.84** | **69.28 ±1.77** | **77.67 ±0.98** | **70.81 ±1.52** | **70.32 ±1.33** | **77.31 ±0.70** | **72.41 ±0.95** |
| Mfeat | BSV | 63.22±6.56 | 60.32±4.60 | 67.42±6.05 | 52.70±4.46 | 48.51±2.18 | 54.19±3.60 | 39.21±4.00 | 33.07±2.87 | 41.68±3.13 |
| | Concat | 75.25±11.82 | 73.15±6.24 | 77.89±8.65 | 57.88±5.73 | 53.11±2.11 | 59.12±4.62 | 41.24±3.71 | 35.48±4.75 | 42.62±3.45 |
| | PVC | 66.80±2.83 | 59.58±1.07 | 68.08±1.89 | 64.26±3.36 | 53.93±2.00 | 65.21±2.21 | 58.19±4.47 | 49.29±3.89 | 59.20±4.21 |
| | IMVC-CBG | 53.50±0.00 | 48.31±0.00 | 53.90±0.00 | 35.20±0.00 | 26.49±0.00 | 35.50±0.00 | 20.95±0.00 | 11.54±0.00 | 21.40±0.00 |
| | PSIMVC-PG | 48.56±0.00 | 45.04±0.00 | 49.96±0.00 | 31.84±0.00 | 25.84±0.00 | 33.34±0.00 | 19.25±0.00 | 10.07±0.00 | 19.55±0.00 |
| | SCSL | 30.55±0.00 | 21.68±0.00 | 33.35±0.00 | 21.10±0.00 | 12.51±0.00 | 24.25±0.00 | 22.50±0.00 | 14.43±0.00 | 26.05±0.00 |
| | **CAMEL** | **99.65 ±0.00** | **99.04 ±0.00** | **99.65 ±0.00** | **96.97 ±6.16** | **97.99 ±2.81** | **97.72 ±4.48** | **96.35 ±6.62** | **96.79 ±6.62** | **97.29 ±4.52** |
| Scene | BSV | 51.18±1.22 | 37.62±0.82 | 54.47±0.98 | 43.09±2.88 | 29.30±1.99 | 45.20±2.63 | 32.65±2.75 | 19.58±1.80 | 34.54±2.32 |
| | Concat | 56.96±4.02 | 44.25±1.45 | 58.04±2.48 | 44.75±1.26 | 29.98±2.59 | 45.48±1.71 | 36.95±2.01 | 22.59±0.87 | 37.54±1.88 |
| | PVC | 55.00±2.40 | 42.46±2.58 | 56.18±2.13 | 46.78±3.36 | 38.14±2.73 | 50.57±3.61 | 42.02±1.06 | 31.75±1.82 | 43.75±0.87 |
| | IMVC-CBG | 42.37±0.00 | 29.10±0.00 | 44.90±0.00 | 27.49±0.00 | 14.77±0.00 | 29.24±0.00 | 20.50±0.00 | 6.19±0.00 | 21.24±0.00 |
| | PSIMVC-PG | 33.82±0.00 | 21.20±0.00 | 35.90±0.00 | 26.90±0.00 | 13.19±0.00 | 28.72±0.00 | 20.19±0.00 | 5.19±0.00 | 20.86±0.00 |
| | SCSL | 48.81±0.00 | 36.90±0.00 | 49.26±0.00 | 16.78±0.00 | 1.87±0.00 | 17.04±0.00 | 19.57±0.00 | 6.65±0.00 | 21.50±0.00 |
| | **CAMEL** | **97.29 ±0.00** | **93.54 ±0.00** | **97.29 ±0.00** | **95.99 ±0.00** | **91.64 ±0.00** | **95.99 ±0.00** | **95.42 ±0.00** | **90.31 ±0.13** | **95.42 ±0.00** |
| Scene15 | BSV | 31.82± 1.44 | 31.59 ±1.01 | 35.64 ±1.22 | 26.00 ±1.57 | 24.53±0.71 | 29.05±0.93 | 22.27±1.27 | 19.31 ±0.67 | 24.69 ±0.91 |
| | Concat | 31.69± 0.58 | 29.67±0.42 | 34.96±0.35 | 27.26 ±0.94 | 25.29 ±0.24 | 31.00 ±0.58 | 21.33±0.69 | 19.75 ± 0.68 | 24.54 ±0.63 |
| | PVC | 30.59± 0.61 | 25.83±0.44 | 33.26±0.68 | 25.71 ±0.80 | 21.01±1.09 | 28.69±0.82 | 24.99±1.31 | 25.35±1.17 | 27.69±1.35 |
| | PSIMVC-PG | 24.82± 0.16 | 21.76±0.20 | 27.62±0.12 | 17.30±0.11 | 11.60±0.19 | 18.42±0.12 | 11.59±0.00 | 3.60±0.00 | 12.01±0.00 |
| | IMVC-CBG | 25.38± 0.17 | 21.82±0.34 | 27.93±0.12 | 17.48±0.12 | 11.57±0.16 | 18.86±0.19 | 11.92±0.00 | 4.05±0.00 | 12.16±0.00 |
| | SCSL | 27.11± 0.00 | 26.77±0.00 | 31.48±0.00 | 21.14±0.00 | 20.22±0.00 | 23.79±0.00 | 12.58±0.00 | 4.25±0.00 | 13.02±0.00 |
| | **CAMEL** | **72.23± 2.02** | **69.10±0.99** | **75.21±1.81** | **61.17±0.97** | **54.00±1.63** | **62.51±1.85** | **57.04±3.30** | **47.33±1.13** | **57.94±1.67** |
| CIFAR10 | BSV | 73.24 ± 5.00 | 64.39 ±2.27 | 75.08 ±3.98 | 59.45 ± 2.28 | 49.20 ± 1.70 | 60.02 ±1.68 | 46.02 ±2.94 | 38.36 ± 1.63 | 47.15 ±2.59 |
| | Concat | 96.14 ±5.84 | 94.87 ±3.87 | 96.17 ±5.78 | 84.09± 10.92 | 81.63 ±8.30 | 84.23 ± 10.77 | 73.31 ±5.72 | 75.43 ± 4.00 | 77.83 ± 5.80 |
| | PVC | 93.00 ± 4.09 | 87.36 ±2.24 | 93.19 ±2.27 | 91.38 ± 2.27 | 83.35 ±1.41 | 91.83 ± 2.27 | 84.55 ± 5.53 | 75.59 ± 4.10 | 84.72 ± 5.36 |
| | PSIMVC-PG | 98.32 ± 0.00 | 96.66 ± 0.00 | 98.32±0.00 | 95.91 ±0.00 | 94.20 ±0.00 | 95.91 ±0.00 | 92.06 ±0.00 | 93.10 ±0.00 | 92.06±0.00 |
| | IMVC-CBG | 98.59 ± 0.00 | 96.61 ± 0.00 | 98.59 ± 0.00 | 96.74 ± 0.00 | 94.17±0.00 | 96.74±0.00 | 94.96 ±0.00 | 92.69 ± 0.00 | 94.96±0.00 |
| | SCSL | OM | OM | OM | OM | OM | OM | OM | OM | OM |
| | **CAMEL** | **99.99 ±0.00** | **99.95 ±0.00** | **99.99 ±0.00** | **99.95 ±0.00** | **99.82 ± 0.00** | **99.95 ±0.00** | **99.46 ±8.10** | **97.20 ±3.26** | **95.89 ±5.46** |
| Noisy MNIST | BSV | 36.68 ± 23.15 | 25.84 ± 23.53 | 37.03 ± 23.42 | 31.29 ± 1.05 | 24.86 ± 0.68 | 35.82 ± 0.84 | 27.62 ± 0.68 | 20.58 ± 0.82 | 31.28 ± 0.78 |
| | Concat | 35.82 ± 1.01 | 31.93 ± 0.62 | 42.19 ± 0.47 | 30.73 ± 1.74 | 24.73 ± 1.39 | 35.55 ± 1.87 | 27.42 ± 1.04 | 20.52 ± 0.77 | 31.20 ± 0.60 |
| | PVC | 39.96 ± 2.99 | 33.93 ± 4.23 | 43.39 ± 3.38 | 31.86 ± 1.98 | 23.21 ± 0.87 | 34.65 ± 1.16 | 35.63 ± 4.24 | 25.68 ± 4.12 | 38.27 ± 3.84 |
| | PSIMVC-PG | 39.99 ± 0.00 | 32.60 ± 0.00 | 41.28 ± 0.00 | 28.39 ± 0.00 | 20.00 ± 0.00 | 30.57 ± 0.00 | 22.44 ± 0.00 | 15.18 ± 0.00 | 25.92 ± 0.00 |
| | IMVC-CBG | 45.67 ± 0.00 | 37.71 ± 0.00 | 46.87 ± 0.00 | 33.87 ± 0.00 | 24.66 ± 0.00 | 34.48 ± 0.00 | 23.75 ± 0.00 | 15.52 ± 0.00 | 26.06 ± 0.00 |
| | SCSL | 21.11 ± 0.00 | 9.68 ± 0.00 | 22.50 ± 0.00 | 16.98 ± 0.00 | 5.86 ± 0.00 | 18.14 ± 0.00 | 14.96 ± 0.00 | 2.40 ± 0.00 | 16.90 ± 0.00 |
| | **CAMEL** | **93.47 ± 7.96** | **95.64 ± 3.32** | **95.36 ± 5.37** | **90.67 ± 7.73** | **94.27 ± 3.24** | **93.31 ± 5.31** | **93.41 ± 7.63** | **95.18 ± 3.16** | **95.31 ± 5.03** |

MSRCV1, Mfeat, Scene, and CIFAR10, most results achieved by CAMEL surpass 95%. Even as the missing rate increases, CAMEL also maintains relatively stable performance.

(2) Compared with both traditional methods (BSV, Concat, and PVC) and recently proposed matrix-based approaches (IMVC-CBG, PSIMVC-PG, and SCSL), CAMEL consistently delivers more competitive results. This superiority stems from the effective integration of connectivity and tensor constraints, which enables a comprehensive modeling of global graph structures, precise learning of latent embeddings, and accurate characterization of high-order correlations across multiple views. In addition, by eliminating the need for post-processing, our approach avoids error accumulation, thereby ensuring robust and high-quality clustering outcomes.

**Runtime comparison:** Table 3 presents a comparison of clustering performance and computational time with and without the use of a constant degree matrix. Specifically, CAMEL-v1 employs a learnable degree matrix, whereas CAMEL adopts a constant degree matrix, with all other components kept identical. A comparative analysis between CAMEL and CAMEL-v1 yields the following two observations:

(1) In several experiments, CAMEL achieves higher values in ACC, NMI, and PUR compared with CAMEL-v1. Even in cases where its performance is slightly lower, the gap remains negligible. This phenomenon can be attributed to the inherent noise in graph structures, which introduces noise into

Table 3: Performance and runtime comparison between CAMEL-v1 and CAMEL.

| Data | Methods | 0.1 | | | | 0.3 | | | | 0.5 | | | |
|---|---|---|---|---|---|---|---|---|---|---|---|---|---|
| | | ACC | NMI | PUR | runtime | ACC | NMI | PUR | runtime | ACC | NMI | PUR | runtime |
| Yale3 | CAMEL-v1 | 66.67±9.67 | 71.97±7.15 | 68.36±9.16 | 0.43 | 71.27±3.39 | 74.71±3.50 | 73.70±3.60 | 0.33 | 64.97±2.76 | 69.86±3.28 | 66.79±3.65 | 0.89 |
| | **CAMEL** | **74.79 ±4.90** | **79.25 ±3.19** | **75.64 ±3.41** | **0.12** | **72.61 ±5.02** | **76.21 ±4.15** | 73.21 ±5.13 | 0.59 | 61.82 ±2.27 | 65.17 ±2.53 | 63.03 ±2.54 | 0.58 |
| MSRCV1 | CAMEL-v1 | 96.86±0.26 | 94.35±0.53 | 96.86±0.26 | 1.04 | 95.52±0.64 | 91.94±0.91 | 95.52±0.64 | 2.78 | 94.57±2.17 | 90.40±2.59 | 94.57±2.17 | 2.78 |
| | **CAMEL** | **97.81 ±0.43** | **95.61 ±0.58** | **97.81 ±0.43** | **0.63** | **97.14 ±0.95** | **94.05 ±1.39** | **97.14 ±0.95** | **0.04** | **97.90 ±0.26** | **96.03 ±0.59** | **97.90 ±0.26** | **0.03** |
| Still-DB | CAMEL-v1 | 54.86±0.53 | 41.66±0.96 | 59.79±0.53 | 1.64 | 62.74±0.50 | 45.69±0.74 | 62.74±0.50 | 8.83 | 71.18±3.84 | 71.55±2.96 | 74.78±3.02 | 8.82 |
| | **CAMEL** | **59.36 ± 1.88** | **45.34 ±1.41** | **61.88 ±1.60** | **0.06** | **67.45 ±0.30** | **53.47 ±0.39** | **67.45 ±0.30** | **0.13** | **74.78 ±0.46** | 66.61 ±0.40 | **74.78 ±0.46** | **0.15** |
| EYaleB10 | CAMEL-v1 | 51.31±0.65 | 47.51±0.30 | 51.31±0.65 | 17.32 | 48.38±3.33 | 46.13±3.58 | 48.66±3.21 | 13.27 | 48.84±2.40 | 43.17±2.83 | 49.16±2.29 | 17.09 |
| | **CAMEL** | **51.97 ±1.78** | **51.09 ±0.98** | **52.44 ±1.58** | **0.56** | **48.59 ±4.73** | **47.11 ±4.73** | **48.94 ±4.49** | **1.38** | **55.72 ±5.42** | **51.21 ±5.48** | **55.78 ±5.35** | **0.97** |
| COIL20MV | CAMEL-v1 | 65.15±2.74 | 77.41±1.08 | 69.00±1.69 | 172.59 | 70.31±4.08 | **81.02±1.69** | 73.63±3.47 | 129.80 | 70.36±1.84 | **79.14±1.42** | 72.53±2.19 | 47.26 |
| | **CAMEL** | **68.38 ±2.27** | **79.17 ±0.98** | **71.22 ±1.84** | **2.38** | 69.28 ±1.77 | 77.67 ±0.98 | 70.81 ±1.52 | 5.06 | 70.32 ±1.33 | 77.31 ±0.70 | 72.41 ±0.95 | 3.51 |
| Mfeat | CAMEL-v1 | **99.80±0.00** | **99.45±0.00** | **99.80±0.00** | 120.82 | 93.77±8.05 | 96.71±3.30 | 95.71±5.40 | 284.37 | **99.33±0.00** | **98.20±0.00** | **99.33±0.00** | 699.72 |
| | **CAMEL** | 99.65 ±0.00 | 99.04 ±0.00 | 99.65 ±0.00 | **5.22** | **96.97 ±6.16** | **97.99 ±2.81** | **97.72 ±4.48** | **0.28** | 96.35 ±6.62 | 96.79 ±6.62 | 97.29 ±4.52 | **4.53** |
| Scene | CAMEL-v1 | **97.46±0.00** | **93.84±0.00** | **97.46±0.00** | 1338.70 | 95.11±0.48 | 90.81±0.62 | 95.11±0.48 | 992.17 | **95.19±0.00** | 89.99±0.11 | **95.19±0.00** | 1336.30 |
| | **CAMEL** | 97.29 ±0.00 | 93.54 ±0.00 | 97.29 ±0.00 | **4.63** | **95.99 ±0.00** | **91.64 ±0.00** | **95.99 ±0.00** | **11.29** | 95.42 ±0.00 | **90.31 ±0.13** | 95.42 ±0.00 | **8.75** |
| Scene15 | CAMEL-v1 | 71.87±1.81 | **71.50±0.78** | **76.30±1.26** | 1686.00 | **66.26±2.31** | **63.50±1.86** | **70.96±1.96** | 4101.80 | **63.14±1.89** | **55.79±1.31** | **65.11±1.23** | 3903.90 |
| | **CAMEL** | **72.23 ±2.02** | 69.10 ±0.99 | 75.21 ±1.81 | **4.36** | 61.17 ±0.97 | 54.00 ±1.63 | 62.51 ±1.85 | **14.57** | 57.04 ±3.30 | 47.33 ±1.13 | 57.94 ±1.67 | **18.71** |
| CIFAR10 | CAMEL-v1 | OM | OM | OM | OM | OM | OM | OM | OM | OM | OM | OM | OM |
| | **CAMEL** | **99.99 ± 0.00** | **99.95±0.00** | **99.99±0.00** | **50.74** | **99.95 ± 0.00** | **99.82 ± 0.00** | **99.95± 0.00** | **281.03** | **93.96±8.10** | **97.20±3.26** | **95.89±5.46** | **244.45** |

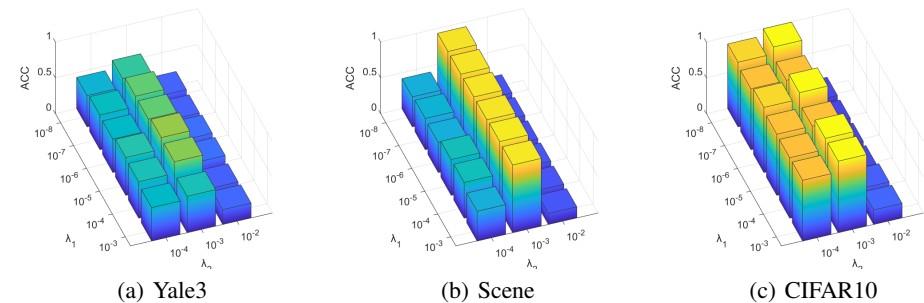

| (a) Yale3 | (b) Scene | (c) CIFAR10 |

Figure 2: Analysis of Hyperparameter Sensitivity in the CAMEL Model

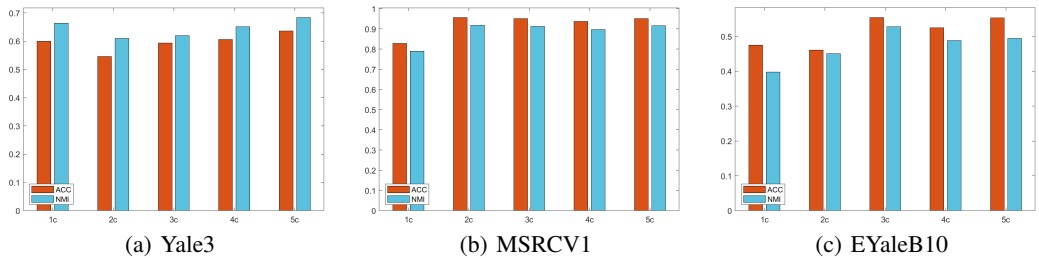

| (a) Yale3 | (b) MSRCV1 | (c) EYaleB10 |

Figure 3: Effect of anchor quantity on the CAMEL model.

the computed degree matrix. Approximating the degree matrix with a suitable constant allows the model to substantially reduce computational complexity without sacrificing clustering accuracy.

(2) On small-scale datasets, the runtime difference between the two strategies is negligible. However, on large-scale datasets, CAMEL runs significantly faster than CAMEL-v1, confirming its lower computational complexity and validating the effectiveness of the constant degree matrix.

### 5.3 PARAMETERS ANALYSIS

**Hyperparameters analysis:** We search for the optimal hyperparameters within the ranges $\lambda_1 \in [10^{-8}, 10^{-3}]$ and $\lambda_2 \in [10^{-4}, 10^{-2}]$. As summarized in Figure 2, the model exhibits slight fluctuations with respect to $\lambda_1$; the best value of $\lambda_2$ is $10^{-3}$, which clearly outperforms the cases of $\lambda_2 = 10^{-2}$ and $\lambda_2 = 10^{-4}$.

**Impact of anchor quantity:** We vary the number of anchors within the range $[c, 5c]$, where $c$ denotes the number of classes. As illustrated in Figure 3, the clustering performance exhibits fluctuations as the anchor count changes. Increasing the number of anchors does not necessarily lead to better results, since additional anchors may introduce more noise while also incurring higher computational cost. Therefore, it is more prudent to select fewer anchors as long as satisfactory per-

formance can be maintained. In practice, setting the anchor number to $2c$ or $3c$ provides a desirable trade-off between performance and efficiency.

**Impact of the degree parameter** $\beta$**:** The parameter $\beta$ is associated with the degree matrix and controls the scaling of node degrees. We search $\beta$ within the range $[10, 10^4]$. As shown in Figure 4, performance also fluctuates with varying $\beta$ values, which can be attributed to instability caused by approximating the degree matrix with a constant. Nevertheless, experimental results suggest that $\beta = 10^3$ and $\beta = 10^4$ are generally reliable choices.

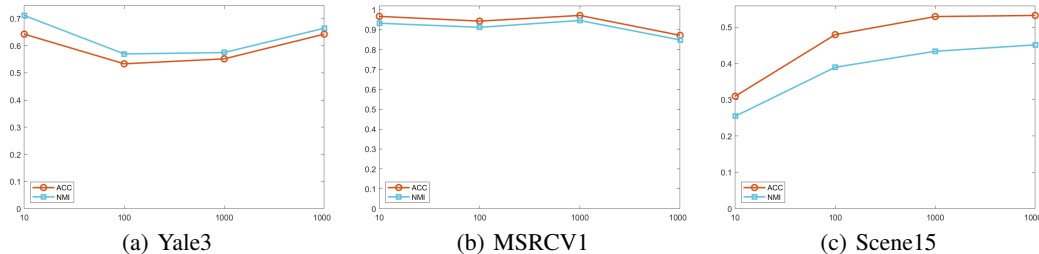

(a) Yale3             (b) MSRCV1             (c) Scene15

Figure 4: Effect of parameter $\beta$ on the CAMEL model.

**Impact of the parameter** $\delta$**:** The parameter $\delta$ belongs to the $\ell_\delta$-norm rank function used in the tensor regularization term. We conduct experiments with $\delta$ ranging from $10^{-3}$ to $1$. As presented in Figure 5, clustering accuracy undergoes significant fluctuations as $\delta$ varies, highlighting the crucial role of tensor regularization in filtering noise. Since the performance is consistently poor when $\delta = 10^{-3}$, we recommend searching for the optimal value within the range $[10^{-2}, 1]$.

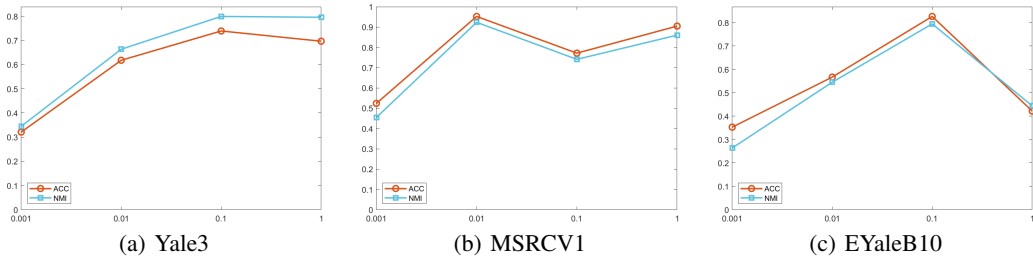

(a) Yale3             (b) MSRCV1             (c) EYaleB10

Figure 5: Effect of parameter $\delta$ on the CAMEL model.

## 5.4 CONVERGENCE

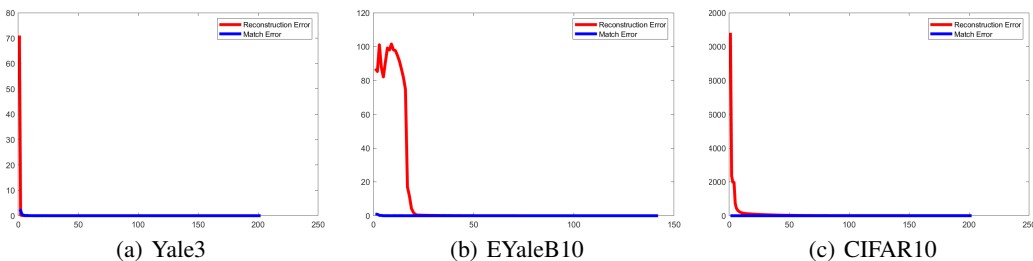

(a) Yale3             (b) EYaleB10             (c) CIFAR10

Figure 6: Training convergence trends of the CAMEL model.

To evaluate convergence, we employ two indicators: the reconstruction error (RE), formulated as $\min_{v} \|\mathbf{X}^v - \mathbf{A}^v \mathbf{Z}^v - \mathbf{E}^v\|_\infty$, and the match error (ME), defined as $\|\mathcal{H} - \mathcal{G}\|_\infty$. As depicted in

Figure 6, both RE and ME decrease rapidly and approach zero within the first 25 iterations, thereby demonstrating strong convergence behavior.

## 5.5 ABLATION STUDY

To investigate the synergistic effect of the connectivity constraint and the tensor constraint, we conducted ablation studies. Specifically, CAMEL-C denotes $\sum_{v=1}^{m} \text{Tr}[(\mathbf{H}^v)^\top \mathbf{L}^v \mathbf{H}^v]$, while CAMEL-T represents $||\mathcal{H}||_{\ell_\delta}$. Table 4 demonstrates that using only CAMEL-C or CAMEL-T yields worse performance than the full model, and the significant performance improvement achieved by combining both components confirms their strong complementarity. Specifically, the connectivity constraint (CAMEL-C) serves as a connected component constraint that explicitly enforces the formation of well-connected clusters in the latent space, effectively preventing the fragmentation of clusters and ensuring that data points from the same category form connected subgraphs. Meanwhile, the tensor constraint (CAMEL-T) operates across multiple views by leveraging the high-order correlations among them, effectively aligning the complementary information from different views into a coherent tensor representation. This dual mechanism allows CAMEL-C to guarantee the structural connectivity of clusters within each view's embedding space, while CAMEL-T captures the global consensus and shared patterns across views, creating a synergistic effect where intra-view connectivity and cross-view consistency mutually enhance each other. Together, they provide a more robust and comprehensive representation learning framework that neither component could achieve alone.

Table 4: Evaluation of the CAMEL model ablation on Yale3, MSRCV1 and EYaleB10 datasets.

| Components | | Yale3 | | | MSRCV1 | | | EYaleB10 | | |
|---|---|---|---|---|---|---|---|---|---|---|
| CAMEL-C | CAMEL-T | ACC | NMI | PUR | ACC | NMI | PUR | ACC | NMI | PUR |
| ✓ | | 28.12 | 28.91 | 30.79 | 41.81 | 29.80 | 42.86 | 25.47 | 16.30 | 26.88 |
| | ✓ | 48.36 | 52.49 | 50.67 | 57.43 | 46.76 | 60.29 | 26.56 | 17.33 | 27.16 |
| ✓ | ✓ | **61.82** | **65.17** | **63.03** | **97.90** | **96.03** | **97.90** | **55.72** | **51.21** | **55.78** |

## 6 CONCLUSION

This paper proposes a unified framework that integrates connectivity constraints with latent embedding tensor constraints. In this framework, view-specific connectivity constraints generate multiple latent embedding matrices, which are jointly optimized in tensor form to capture high-order correlations across views. Cluster labels are directly obtained by applying $k$-means on the learned embeddings, thereby avoiding additional post-processing. To reduce the computational complexity from $\mathcal{O}(n^2)$ to $\mathcal{O}(1)$, we introduce a constant-degree matrix into the Laplacian construction of the connectivity constraint. Extensive experiments demonstrate the effectiveness and efficiency of CAMEL, while relatively high standard deviations on certain datasets reveal a limitation of the constant-degree approximation.

## ACKNOWLEDGEMENTS

This work was supported in part by the National Natural Science Foundation of China (No. 62506116, No. U24A20331), in part by the Hebei Natural Science Foundation (No. F2025205002), in part by the Science Research Project of the Hebei Education Department (No. BJ2026004), and in part by the Beijing Natural Science Foundation (No. L251005).

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

# A  APPENDIX

The detailed optimization derivations, the proof of convergence, and the remaining experimental results are provided in this appendix.

## A.1  LLM USAGE

We employ LLMs solely to refine the language and wording for improved fluency and precision, with no further application beyond this purpose.

## A.2  DETAILED OPTIMIZATION PROCESS

The augmented Lagrangian function of the optimization problem is given as:

$$
\begin{aligned}
&\mathcal{L}(\{\mathbf{A}^v\}_{n=1}^m, \{\mathbf{E}^v\}_{v=1}^m, \{\mathbf{Z}^v\}_{v=1}^m, \{\mathbf{H}^v\}_{v=1}^m, \{\mathbf{Y}^v\}_{v=1}^m, \{\mathbf{B}^v\}_{v=1}^m, \mathcal{W}, \mathcal{G}) \\
&= \|\mathcal{G}\|_{\ell_\delta} + \lambda_1 \sum_{v=1}^m \|\mathbf{E}^v\|_F^2 + \lambda_2 \sum_{v=1}^m \mathrm{Tr}[(\mathbf{H}^v)^\top (\mathbf{I} - \frac{1}{\beta}(\mathbf{Z}^v)^\top \mathbf{Z}^v)\mathbf{H}^v] + \langle \mathcal{W}, \mathcal{H} - \mathcal{G} \rangle \\
&+ \frac{\zeta}{2}\|\mathcal{H} - \mathcal{G}\|_F^2 + \sum_{v=1}^m \left( \langle \mathbf{Y}^v, \mathbf{X}^v - \mathbf{A}^v \mathbf{Z}^v - \mathbf{E}^v \rangle + \frac{\mu}{2}\|\mathbf{X}^v - \mathbf{A}^v \mathbf{Z}^v - \mathbf{E}^v\|_F^2 \right)
\end{aligned}
\tag{5}
$$

We can decompose Eq. 5 into several subproblems and solve them sequentially.

**$\mathbf{A}^v$-Subproblem**  By fixing the other variables, the update of $\mathbf{A}^v$ can be obtained from the following optimization:

$$
\mathbf{A}^v = \underset{(\mathbf{A}^v)^T \mathbf{A}^v = \mathbf{I}}{arg\ max}\ \mathrm{Tr}((\mathbf{A}^v)^T \mathbf{M}^v)
\tag{6}
$$

where $\mathbf{M}^v = (\mathbf{Y}^v + \mu \mathbf{X}^v - \mu \mathbf{E}^v)(\mathbf{Z}^v)^T$. The optimal solution of $\mathbf{A}^v$ is $\mathbf{U}_{\mathbf{A}^v}\mathbf{V}_{\mathbf{A}^v}^T$, where $\mathbf{U}_{\mathbf{A}^v}$ and $\mathbf{V}_{\mathbf{A}^v}$ are the left and right singular matrix of $\mathbf{M}^v$.

**$\mathbf{H}^v$-Subproblem**  Treating the other variables as constants, the subproblem with respect to $\mathbf{H}^v$ is formulated as:

$$
\min_{(\mathbf{H}^v)^\top \mathbf{H}^v = \mathbf{I}} \lambda_2 Tr((\mathbf{H}^v)^\top \mathbf{L}^v \mathbf{H}^v) + \mathrm{Tr}((\mathbf{W}^v)^\top(\mathbf{H}^v - \mathbf{G}^v)) + \frac{\zeta}{2}\|\mathbf{H}^v - \mathbf{G}^v\|_F^2
\tag{7}
$$

This subproblem can be equivalently reformulated as:

$$
\min_{(\mathbf{H}^v)^\top \mathbf{H}^v = \mathbf{I}} \mathrm{Tr}((\mathbf{H}^v)^\top (\lambda_2 \mathbf{L}^v)\mathbf{H}^v) + \mathrm{Tr}((\mathbf{H}^v)^\top (\mathbf{W}^v - \zeta \mathbf{G}^v))
\tag{8}
$$

To make the problem more tractable, we further relax it into the following form:

$$
\max_{(\mathbf{H}^v)^\top \mathbf{H}^v = \mathbf{I}} \mathrm{Tr}((\mathbf{H}^v)^\top \widetilde{\mathbf{L}}^v \mathbf{H}^v) + \mathrm{Tr}((\mathbf{H}^v)^\top \mathbf{C})
\tag{9}
$$

where $\widetilde{\mathbf{L}}^v = \alpha \mathbf{I} - \lambda_2 \mathbf{L}^v$, $\mathbf{C} = \zeta \mathbf{G}^v - \mathbf{W}^v$. The constant parameter $\alpha$ is introduced to ensure that $\widetilde{\mathbf{L}}^v$ is positive definite, and it is fixed at 0.001 throughout all experiments. It is clear that this becomes a quadratic optimization problem on the Stiefel manifold Nie et al. (2017), which can be efficiently solved by Algorithm 1. Let $\mathbf{D} = \beta \mathbf{I}$. In this case, we have $\mathbf{D}^{-\frac{1}{2}} = \frac{1}{\sqrt{\beta}}\mathbf{I}$. Consequently, $\widetilde{\mathbf{L}}^v$ can be expressed as $\widetilde{\mathbf{L}}^v = \alpha \mathbf{I} - \lambda_2 \mathbf{L}^v = (\alpha - \lambda_2)\mathbf{I} + \frac{\lambda_2}{\beta}(\mathbf{Z}^v)^\top \mathbf{Z}^v$. In Algorithm 1, the adjacency matrix $(\mathbf{Z}^v)^\top \mathbf{Z}^v \in \mathbb{R}^{n \times n}$ is multiplied by $\mathbf{H}^v \in \mathbb{R}^{n \times c}$. To improve efficiency, the computation is reordered as $(\mathbf{Z}^v)^\top (\mathbf{Z}^v \mathbf{H}^v)$, which reduces the complexity to $\mathcal{O}(tcn)$ and avoids the $\mathcal{O}(n^2)$ cost of explicitly forming $(\mathbf{Z}^v)^\top \mathbf{Z}^v$.

**$\mathbf{E}^v$-Subproblem**  By fixing the other variables, the subproblem with respect to $\mathbf{E}^v$ can be formulated as:

$$
\min_{\mathbf{E}^v} \lambda_1 \|\mathbf{E}^v\|_F^2 + \langle \mathbf{Y}^v, \mathbf{X}^v - \mathbf{A}^v \mathbf{Z}^v - \mathbf{E}^v \rangle + \frac{\mu}{2}\|\mathbf{X}^v - \mathbf{A}^v \mathbf{Z}^v - \mathbf{E}^v\|_F^2
\tag{10}
$$

Minimizing this function with respect to $\mathbf{E}^v$ yields the following optimal solution:

$$
\mathbf{E}^v = \frac{1}{2\lambda_1 + \mu}[\mathbf{Y}^v + \mu(\mathbf{X}^v - \mathbf{A}^v \mathbf{Z}^v)]
\tag{11}
$$

---

**Algorithm 1** Optimization Algorithm of $\mathbf{H}^v$-Subproblem

---

**Input:** $\mathbf{X}^v$,$\mathbf{H}^v$,$C$
**Output:** $\mathbf{H}^v$

1: ensure $(\mathbf{H}^v)^\top \mathbf{H}^v = \mathbf{I}; \widetilde{\mathbf{L}}^v = \alpha \mathbf{I} - \lambda_2 \mathbf{L}^v$ is a positive definite matrix.
2: **while** not converge **do**
3:     Update $\mathbf{F} = 2\widetilde{\mathbf{L}}^v \mathbf{H}^v + \mathbf{C} = 2(\alpha - \lambda_2)\mathbf{H}^v + \frac{2\lambda_2}{\beta}(\mathbf{Z}^v)^\top (\mathbf{Z}^v \mathbf{H}^v) + \mathbf{C}$
4:     Compute the SVD of matrix F:$\mathbf{U}\mathbf{\Sigma}\mathbf{V}^\top = \mathbf{F}$
5:     Update $\mathbf{H}^v = \mathbf{U}\mathbf{V}^\top$
6: **end while**
7: Output $\mathbf{H}^v$

---

$\mathbf{Z}^v$**-Subproblem**  When focusing on $\mathbf{Z}^v$ while treating the remaining variables as fixed, the optimization problem can be expressed as:

$$\min_{\mathbf{Z}^v \geq 0} \left( \langle \mathbf{Y}^v, \mathbf{X}^v - \mathbf{A}^v \mathbf{Z}^v - \mathbf{E}^v \rangle + \frac{\mu}{2}\|\mathbf{X}^v - \mathbf{A}^v \mathbf{Z}^v - \mathbf{E}^v\|_F^2 \right) \tag{12}$$

The above formulation imposes a nonnegativity constraint, making $\mathbf{Z}^v$ essentially a nonnegative representation matrix to be optimized. Accordingly, the update rule for $\mathbf{Z}^v$ is:

$$\mathbf{Z}^v = \max(0, \frac{1}{\mu}\left[(\mathbf{A}^v)^\top \mathbf{Y}^v + \mu(\mathbf{A}^v)^\top (\mathbf{X}^v - \mathbf{E}^v)\right]) \tag{13}$$

$\mathcal{G}$**-Subproblem**  Keeping other variables fixed, $\mathcal{G}$ can be derived by solving

$$\mathcal{G} = \arg\min_{\mathcal{G}} \frac{1}{\zeta}\|\mathcal{G}\|_{\ell_\delta} + \frac{1}{2}\|\mathcal{G} - (\mathcal{H} + \frac{\mathcal{W}}{\zeta})\|_F^2 \tag{14}$$

By applying the following theorem, the optimal solution of $\mathcal{G}$ can be determined:

**Theorem 2** *Suppose $\mathcal{D} \in \mathbb{R}^{n_1 \times n_2 \times n_3}$ is a tensor, and let its tensor singular value decomposition (t-SVD) be $\mathcal{D} = \mathcal{U} * \mathcal{S} * \mathcal{V}^\top$. Then the minimization problem under the $\ell_\delta$-norm can be formulated as follows:*

$$\min_{\mathcal{G}} \zeta\|\mathcal{G}\|_{\ell_\delta} + \frac{1}{2}\|\mathcal{G} - \mathcal{D}\|_F^2 \tag{15}$$

*The closed-form optimal solution to Eq. (15) can be expressed as:*

$$\mathcal{G}^* = \mathcal{U} * ifft(\Omega_{f,\zeta}(\mathcal{S}_f), [], 3) * \mathcal{V}^\top \tag{16}$$

*Here, $ifft(\Omega_{f,\zeta}(\mathcal{S}_f), [], 3)$ represents a tensor with diagonal frontal slices. For each slice, the diagonal entry $\Omega_{f,\zeta}(\mathcal{S}_f^k(i,i))$ is determined by solving the following subproblem:*

$$\Omega_{f,\zeta}(\mathcal{S}_f^k(i,i)) = \arg\min_{x \geq 0} \frac{1}{2}\left(x - \mathcal{S}_f^k(i,i)\right)^2 + \zeta f(x) \tag{17}$$

*where the parameter satisfies $\zeta > 0$, and $f(x)$ represents the rank function of the $\ell_\delta$-norm, namely $f(x) = \frac{(1+\delta)x}{\delta+x}$.*

Since Eq. (17) involves a mixture of convex and concave components, it can be solved using the framework of difference-of-convex programming Tao & An (1997), resulting in the closed-form expression below:

$$\kappa^{iter+1} = \left(\mathcal{S}_f^k(i,i) - \frac{\partial f(\kappa^{iter})}{\zeta}\right)_+ \tag{18}$$

where $\kappa = \Omega_{f,\zeta}(\mathcal{S}_f^k(i,i))$ and $iter$ refers to the iteration step.

**Lagrange multipliers and penalty parameters**  The penalty parameters $\mu$ and $\zeta$ together with the Lagrange multipliers $\mathcal{W}$ and $\mathbf{Y}^v$ are updated according to:

$$\begin{cases} \mathbf{Y}^v = \mathbf{Y}^v + \mu(\mathbf{X}^v - \mathbf{A}^v\mathbf{Z}^v - \mathbf{E}^v) \\ \mathcal{W} = \mathcal{W} + \zeta(\mathcal{H} - \mathcal{G}) \\ \mu = \eta_\mu\mu, \mu = min(\mu, \mu_{max}) \\ \zeta = \eta_\zeta\zeta, \zeta = min(\zeta, \zeta_{max}) \end{cases} \tag{19}$$

**Impute $\mathbf{X}^v$**

$$\mathbf{X}^v_{:,mis} = (\mathbf{A}^v\mathbf{Z}^v)_{:,mis} \tag{20}$$

This step performs imputation for the missing entries in the original data, where the subscript $mis$ denotes the indices of the missing samples. A complete description of the optimization process for CAMEL is summarized in Algorithm 2, where the subscript $ava$ refers to the indices of the available data points.

---

**Algorithm 2** Optimization Algorithm of CAMEL

---

**Input:** Incomplete multi-view dataset $\{\mathbf{X}^v\}_{v=1}^m$, cluster count $c$, balancing parameters $\lambda_1, \lambda_2$ and anchor number $t$.
**Output:** Clustering labels
1: **Initialize:** $\forall v, (\mathbf{Z}^v)_{:,mis} = \mathbf{1}, (\mathbf{Z}^v)_{:,ava} = \mathbf{0}, \mathbf{H}^v = \mathbf{0}, \mathbf{A}^v = \mathbf{0}, \mathbf{E}^v = \mathbf{0}, \mathbf{Y}^v = \mathbf{0}, \mathcal{G} = \mathbf{0}, \mathcal{W} = \mathbf{0}, \mu = 10^{-4}, \zeta = 10^{-4}, \eta_\mu = \eta_\zeta = 1.2, \mu_{\max} = \zeta_{\max} = 10^{12}, \epsilon = 10^{-5}$
2: **while** not converge **do**
3:     Update $\{\mathbf{A}^v\}_{v=1}^m$ by Eq. (6)
4:     Update $\{\mathbf{E}^v\}_{v=1}^m$ by Eq. (11)
5:     Update $\{\mathbf{Z}^v\}_{v=1}^m$ by Eq. (13)
6:     Update $\{\mathbf{H}^v\}_{v=1}^m$ by Algorithm 1
7:     Update $\{\mathbf{Y}^v\}_{v=1}^m, \mathcal{W}, \mu, \zeta$ by Eq. (19)
8:     Update $\{\mathcal{G}^v\}_{v=1}^m$ by Eq. (16)
9:     Update $\{\mathbf{X}^v\}_{v=1}^m$ by Eq. (20)
10:     Check the convergence conditions: $\|\mathbf{X}^v - \mathbf{A}^v\mathbf{Z}^v - \mathbf{E}^v\|_\infty < \epsilon$ and $\|\mathbf{H}^v - \mathcal{G}^v\|_\infty < \epsilon$
11: **end while**
12: The final clustering assignments are generated via $k$-means applied to the concatenated latent embedding matrix $[\mathbf{H}^1, \mathbf{H}^2, \cdots, \mathbf{H}^m]$.

---

### A.3 CONVERGENCE ANALYSIS

This subsection provides a detailed proof showing that Theorem 1 in Section 4.2 establishes the theoretical guarantee of CAMEL's convergence. We first introduce a necessary lemma:

**Lemma 1** *Lewis & Sendov (2005) The expression $F(\mathbf{X}) = f(\boldsymbol{v}(\mathbf{X}))$ defines a mapping $F : \mathbb{R}^{m \times n} \to \mathbb{R}$, where $\boldsymbol{v}(\mathbf{X}) = (\sigma_1(\mathbf{X}), \ldots, \sigma_r(\mathbf{X}))$ represents the collection of singular values of $\mathbf{X} \in \mathbb{R}^{m \times n}$, with $r = \min(m, n)$. The singular value decomposition of $\mathbf{X}$ is written as $\mathbf{X} = \mathbf{U}\mathrm{diag}(\boldsymbol{v}(\mathbf{X}))\mathbf{V}^\top$. Moreover, assume that the function $f : \mathbb{R}^r \to \mathbb{R}$ is both absolutely symmetric and differentiable at the point $\boldsymbol{v}(\mathbf{X})$. With these conditions in place, the subdifferential of $F(\mathbf{X})$ at $\mathbf{X}$ can be expressed as*

$$\frac{\partial F(\mathbf{X})}{\partial \mathbf{X}} = \mathbf{U}\mathrm{diag}(\partial f(\boldsymbol{v}(\mathbf{X})))\mathbf{V}^\top \tag{21}$$

*where $\partial f(\boldsymbol{v}(\mathbf{X})) = \left( \frac{\partial f(\sigma_1(\mathbf{X}))}{\partial \mathbf{X}}, \ldots, \frac{\partial f(\sigma_r(\mathbf{X}))}{\partial \mathbf{X}} \right)$.*

**Proof that the sequence $\{\mathcal{J}_p\}_{p=1}^\infty$ is bounded:**  Consider the $(p + 1)$-th iteration. Solving for $\mathbf{E}^v_{q,p+1}$ columnwise yields:

$$\mathbf{e}^v_{q,p+1} = \frac{1}{2\lambda_1 + \mu_p}\Big(\mu_p(\mathbf{x}^v_q - \mathbf{A}^v\mathbf{z}^v_{q,p+1}) + \mathbf{y}^v_{q,p}\Big). \tag{22}$$

Moreover, the multiplier $\mathbf{Y}^v$ update is given by:
$$\mathbf{Y}_{p+1}^v = \mathbf{Y}_p^v + \mu_p(\mathbf{X}^v - \mathbf{A}^v\mathbf{Z}_{p+1}^v - \mathbf{E}_{p+1}^v) \tag{23}$$
The $q$-th column $\mathbf{y}_{q,p+1}^v$ of the multiplier $\mathbf{Y}^v$ is updated as:
$$\mathbf{y}_{q,p+1}^v = \mathbf{y}_{q,p}^v + \mu_p(\mathbf{x}_q^v - \mathbf{A}^v\mathbf{z}_{q,p+1}^v - \mathbf{e}_{q,p+1}^v) \tag{24}$$

Substituting Eq.22 into the Eq.24 gives:
$$\mathbf{y}_{q,p+1}^v = \frac{2\lambda_1}{2\lambda_1 + \mu_p}\mathbf{y}_{q,p}^v + \frac{2\lambda_1\mu_p}{2\lambda_1 + \mu_p}(\mathbf{x}_q^v - \mathbf{A}^v\mathbf{z}_{q,p+1}^v) \tag{25}$$

Taking the $\ell_2$-norm of both sides yields the recursive inequality:
$$\|\mathbf{y}_{q,p+1}^v\|_2 \leq \underbrace{\frac{2\lambda_1}{2\lambda_1 + \mu_p}}_{\iota_p} \|\mathbf{y}_{q,p}^v\|_2 + \underbrace{\frac{2\lambda_1\mu_p}{2\lambda_1 + \mu_p}}_{\gamma_p} \|\mathbf{x}_q^v - \mathbf{A}^v\mathbf{z}_{q,p+1}^v\|_2 \tag{26}$$

where $\iota_p, \gamma_p > 0$ are bounded constants since $\lambda_1$ is constant and $\mu_p$ is bounded. Finally, given that the initial multiplier $\mathbf{y}_{q,0}^v$, the penalty parameters $\{\mu_p\}$, and the data term $\mathbf{x}_q^v - \mathbf{A}^v\mathbf{z}_{q,p}^v$ are bounded, the recursive inequality implies $\sup_p \|\mathbf{y}_{q,p}^v\|_2 < \infty$, and thus the sequence $\{\mathbf{Y}_p^v\}$ is bounded.

At iteration $(p+1)$, the first-order optimality condition regarding $\boldsymbol{\mathcal{G}}_{p+1}$ is characterized by:
$$\mathbf{0} = \partial\|\boldsymbol{\mathcal{G}}_{p+1}\|_{\ell_\delta} + \zeta_p(\boldsymbol{\mathcal{G}}_{p+1} - \boldsymbol{\mathcal{H}}_{p+1}) - \boldsymbol{\mathcal{W}}_p \tag{27}$$
Meanwhile, the variable $\boldsymbol{\mathcal{W}}$ is updated as:
$$\boldsymbol{\mathcal{W}}_{p+1} = \boldsymbol{\mathcal{W}}_p + \zeta_p(\boldsymbol{\mathcal{H}}_{p+1} - \boldsymbol{\mathcal{G}}_{p+1}) \tag{28}$$
By combining these two relations, we arrive at:
$$\partial\|\boldsymbol{\mathcal{G}}_{p+1}\|_{\ell_\delta} = \boldsymbol{\mathcal{W}}_{p+1} \tag{29}$$
The tensor $\boldsymbol{\mathcal{G}}$ can be factorized via the t-SVD as $\boldsymbol{\mathcal{G}} = \boldsymbol{\mathcal{U}} * \boldsymbol{\mathcal{S}} * \boldsymbol{\mathcal{V}}^\top$. Applying Lemma 1, we obtain:
$$\left\|\partial\|\boldsymbol{\mathcal{G}}_{p+1}\|_{\ell_\delta}\right\|_F^2 = \left\|\frac{1}{n_3}\boldsymbol{\mathcal{U}} * \text{ifft}\left(\partial f\left(\boldsymbol{\mathcal{S}}_f\right), [], 3\right) * \boldsymbol{\mathcal{V}}^\top\right\|_F^2$$
$$= \frac{1}{n_3^3}\|\partial f(\boldsymbol{\mathcal{S}}_f)\|_F^2 \leq \frac{1}{n_3^3}\sum_{k=1}^{n_3}\sum_{i=1}^{\min(n_1,n_2)}[\partial f(\boldsymbol{\mathcal{S}}_f^k(i,i))]^2 \tag{30}$$

As a consequence, the Frobenius measure of $\partial\|\boldsymbol{\mathcal{G}}_{p+1}\|_{\ell_\delta}$ is controlled by a finite constant, which ensures its boundedness. Combining this fact with Eq. (29), it follows that the sequence $\{\boldsymbol{\mathcal{W}}_p\}_{p=1}^\infty$ is uniformly bounded. In addition, making use of the update scheme outlined in Algorithm 2, one can establish the inequality below:
$$\begin{aligned}
&\mathcal{L}(\mathbf{A}_{p+1}^v, \mathbf{E}_{p+1}^v, \mathbf{Z}_{p+1}^v, \mathbf{H}_{p+1}^v, \boldsymbol{\mathcal{G}}_{p+1}, \mathbf{Y}_p^v, \boldsymbol{\mathcal{W}}_p, \mu_p, \zeta_p) \\
&\leq \mathcal{L}(\mathbf{A}_p^v, \mathbf{E}_p^v, \mathbf{Z}_p^v, \mathbf{H}_p^v, \boldsymbol{\mathcal{G}}_p, \mathbf{Y}_p^v, \boldsymbol{\mathcal{W}}_p, \mu_p, \zeta_p) \\
&= \mathcal{L}(\mathbf{A}_p^v, \mathbf{E}_p^v, \mathbf{Z}_p^v, \mathbf{H}_p^v, \boldsymbol{\mathcal{G}}_p, \mathbf{Y}_{p-1}^v, \boldsymbol{\mathcal{W}}_{p-1}, \mu_{p-1}, \zeta_{p-1}) \\
&\quad + \frac{\zeta_p + \zeta_{p-1}}{2\zeta_{p-1}^2}\|\boldsymbol{\mathcal{W}}_p - \boldsymbol{\mathcal{W}}_{p-1}\|_F^2 \\
&\quad + \frac{\mu_p + \mu_{p-1}}{2\mu_{p-1}^2}\sum_{v=1}^m\|\mathbf{Y}_p^v - \mathbf{Y}_{p-1}^v\|_F^2
\end{aligned} \tag{31}$$
Iterating the inequality over $p = 1, 2, \ldots, n$ yields:
$$\begin{aligned}
&\mathcal{L}(\mathbf{A}_{p+1}^v, \mathbf{E}_{p+1}^v, \mathbf{Z}_{p+1}^v, \mathbf{H}_{p+1}^v, \boldsymbol{\mathcal{G}}_{p+1}, \mathbf{Y}_p^v, \boldsymbol{\mathcal{W}}_p, \mu_p, \zeta_p) \\
&\leq \mathcal{L}(\mathbf{A}_1^v, \mathbf{E}_1^v, \mathbf{Z}_1^v, \mathbf{H}_1^v, \boldsymbol{\mathcal{G}}_1, \mathbf{Y}_0^v, \boldsymbol{\mathcal{W}}_0, \mu_0, \zeta_0) \\
&\quad + \sum_{p=1}^n \frac{\zeta_p + \zeta_{p-1}}{2\zeta_{p-1}^2}\|\boldsymbol{\mathcal{W}}_p - \boldsymbol{\mathcal{W}}_{p-1}\|_F^2 \\
&\quad + \sum_{p=1}^n \left(\frac{\mu_p + \mu_{p-1}}{2\mu_{p-1}^2}\sum_{v=1}^m\|\mathbf{Y}_p^v - \mathbf{Y}_{p-1}^v\|_F^2\right)
\end{aligned} \tag{32}$$

It follows directly that:

$$\sum_{p=1}^{n} \frac{\zeta_p + \zeta_{p-1}}{2\zeta_{p-1}^2} < \infty, \quad \sum_{p=1}^{n} \frac{\mu_p + \mu_{p-1}}{2\mu_{p-1}^2} < \infty \tag{33}$$

Since the initial value of the objective function $\mathcal{L}(\mathbf{A}_1^v, \mathbf{E}_1^v, \mathbf{Z}_1^v, \mathbf{H}_1^v, \mathcal{G}_1, \mathbf{Y}_0^v, \mathcal{W}_0, \mu_0, \zeta_0)$ is finite, and the sequences $\{\mathbf{Y}_p^v\}_{p=1}^{\infty}$, $\{\mathcal{W}_p\}_{p=1}^{\infty}$ together with the sums $\sum_{p=1}^{n} \frac{\zeta_p + \zeta_{p-1}}{2\zeta_{p-1}^2}$ and $\sum_{p=1}^{n} \frac{\mu_p + \mu_{p-1}}{2\mu_{p-1}^2}$ are uniformly bounded, it follows that the augmented Lagrangian $\mathcal{L}(\mathbf{A}_{p+1}^v, \mathbf{E}_{p+1}^v, \mathbf{Z}_{p+1}^v, \mathbf{H}_{p+1}^v, \mathcal{G}_{p+1}, \mathbf{Y}_p^v, \mathcal{W}_p, \mu_p, \zeta_p)$ is bounded throughout the entire iterative process.

We next make use of the following identity:

$$\mathcal{L}(\mathbf{A}_{p+1}^v, \mathbf{E}_{p+1}^v, \mathbf{Z}_{p+1}^v, \mathbf{H}_{p+1}^v, \mathcal{G}_{p+1}, \mathbf{Y}_p^v, \mathcal{W}_p, \mu_p, \zeta_p)$$

$$= \|\mathcal{G}_{p+1}\|_{\ell_\delta} + \lambda_1 \sum_{v=1}^{m} \|\mathbf{E}_{p+1}^v\|_F^2 + \lambda_2 \sum_{v=1}^{m} \mathrm{Tr}[(\mathbf{H}_{p+1}^v)^\top (\mathbf{I} - \frac{1}{\beta}(\mathbf{Z}_{p+1}^v)^\top \mathbf{Z}_{p+1}^v)\mathbf{H}_{p+1}^v] + \langle \mathcal{W}_p, \mathcal{H}_{p+1} - \mathcal{G}_{p+1} \rangle$$

$$+ \frac{\zeta_p}{2}\|\mathcal{H}_{p+1} - \mathcal{G}_{p+1}\|_F^2 + \sum_{v=1}^{m}\left( \langle \mathbf{Y}_p^v, \mathbf{X}^v - \mathbf{A}_{p+1}^v\mathbf{Z}_{p+1}^v - \mathbf{E}_{p+1}^v \rangle + \frac{\mu_p}{2}\|\mathbf{X}^v - \mathbf{A}_{p+1}^v\mathbf{Z}_{p+1}^v - \mathbf{E}_{p+1}^v\|_F^2 \right) \tag{34}$$

All quantities on the right-hand side of Eq. (34) are finite. Among them, the term $\|\mathcal{G}_{p+1}\|_{\ell_\delta}$ is of particular importance: its boundedness ensures that the corresponding singular values $\mathcal{S}_f^k(i,i)$ do not diverge. This observation leads directly to the relation below:

$$\|\mathcal{G}_{p+1}\|_F^2 = \frac{1}{n_3}\|\mathcal{G}_{f,p+1}\|_F^2 = \frac{1}{n_3}\sum_{k=1}^{n_3}\sum_{i=1}^{\min(n_1,n_2)}[\mathcal{S}_f^k(i,i)]^2 \tag{35}$$

This guarantees that the sequence $\{\mathcal{G}_p\}_{p=1}^{\infty}$ remains bounded. In addition, the update rules immediately show that $\{\mathbf{A}_p^v\}_{p=1}^{\infty}$, $\{\mathbf{H}_p^v\}_{p=1}^{\infty}$, $\{\mathbf{Z}_p^v\}_{p=1}^{\infty}$, and $\{\mathbf{E}_p^v\}_{p=1}^{\infty}$ are bounded sequences as well. Consequently, the combined sequence $\left\{\mathcal{J}_p = \mathbf{A}_p^v, \mathbf{E}_p^v, \mathbf{Z}_p^v, \mathbf{H}_p^v, \mathbf{Y}_p^v, \mathcal{G}_p, \mathcal{W}_p\right\}_{p=1}^{\infty}$ is confined to a bounded set.

**On the convergence of accumulation points towards stationary KKT solutions:** According to the Weierstrass–Bolzano theorem Bartle & Sherbert (2000), any bounded sequence admits a convergent subsequence. Hence, $\{\mathcal{J}_p\}_{p=1}^{\infty}$ has at least one accumulation point, denoted by $\mathcal{J}_*$. Consequently, we obtain:

$$\lim_{p\to\infty}(\mathbf{A}_p^v, \mathbf{E}_p^v, \mathbf{Z}_p^v, \mathbf{H}_p^v, \mathbf{Y}_p^v, \mathcal{G}_p, \mathcal{W}_p) = (\mathbf{A}_*^v, \mathbf{E}_*^v, \mathbf{Z}_*^v, \mathbf{H}_*^v, \mathbf{Y}_*^v, \mathcal{G}_*, \mathcal{W}_*) \tag{36}$$

From Eq. (19), the following relations can be derived:

$$\mathbf{X}^v - \mathbf{A}_{p+1}^v\mathbf{Z}_{p+1}^v - \mathbf{E}_{p+1}^v = \frac{\mathbf{Y}_{p+1}^v - \mathbf{Y}_p^v}{\mu_p}, \mathbf{\mathcal{Z}}_{p+1} - \mathcal{G}_{p+1} = \frac{\mathcal{W}_{p+1} - \mathcal{W}_p}{\zeta_p} \tag{37}$$

Since the sequences $\{\mathbf{Y}_p^v\}_{p=1}^{\infty}$ and $\{\mathcal{W}_p\}_{p=1}^{\infty}$ remain bounded, the accumulation point satisfies the following conditions:

$$\mathbf{X}^v - \mathbf{A}_*^v\mathbf{Z}_*^v - \mathbf{E}_*^v = \mathbf{0}, \mathcal{H}_* - \mathcal{G}_* = \mathbf{0} \tag{38}$$

Moreover, the satisfaction of the first-order optimality conditions by $\mathbf{E}_{p+1}^v$ and $\mathcal{G}_{p+1}$ leads to:

$$\mathbf{Y}_*^v = \lambda_1 \partial\|\mathbf{E}_*^v\|_F^2, \quad \mathcal{W}_* = \partial\|\mathcal{G}_*\|_{\ell_\delta} \tag{39}$$

Thus, we deduce that the accumulation point $\mathcal{J}_*$ complies with both stationarity and primal feasibility. Accordingly, any subsequential limit of $\{\mathcal{J}_p\}_{p=1}^{\infty}$ constitutes a stationary point obeying the KKT conditions of the proposed optimization model.

### A.4 SUPPLEMENTARY EXPERIMENTS

This subsection supplements the experimental results that are not included in the main text. Tables 5 and 6 present the remaining ablation studies, which further validate the synergistic effect of the connectivity constraint and the tensor constraint in CAMEL. Figures 7, 8, 9, and 10 illustrate the influence of different parameters on other datasets, namely hyperparameters, the number of anchors $t$, the degree matrix parameter $\beta$, and the rank function parameter $\delta$. Figure 11 illustrates the convergence behavior on the remaining datasets, again demonstrating the favorable convergence of CAMEL.

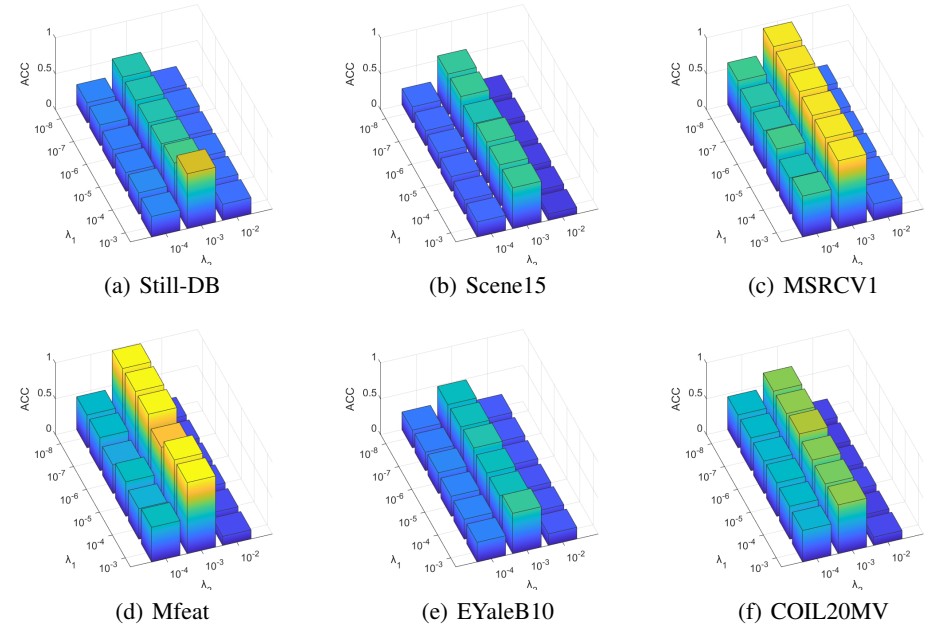

Figure 7: Analysis of Hyperparameter Sensitivity in the CAMEL Model on the rest datasets.

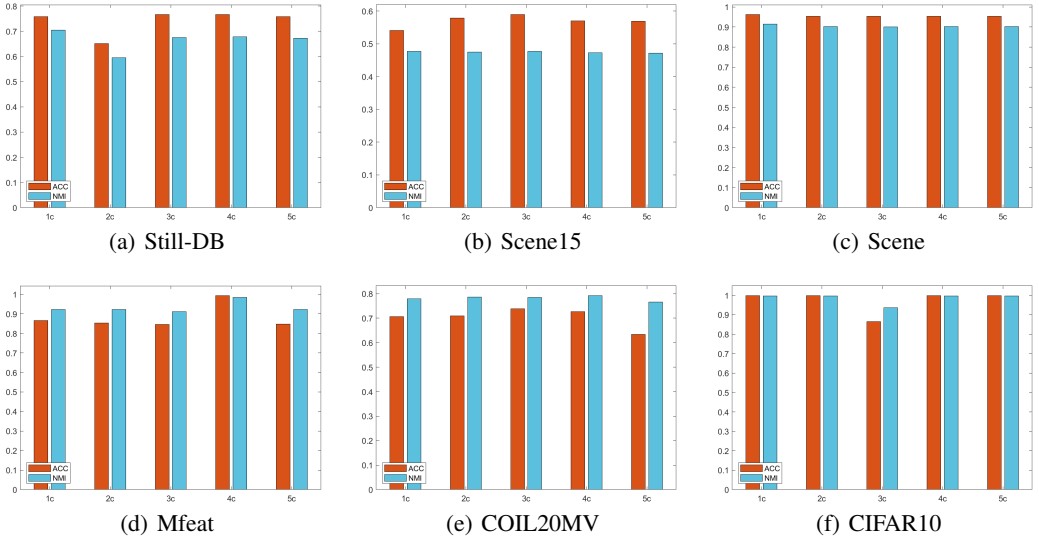

Figure 8: Effect of anchor quantity on the CAMEL model on the rest datasets.

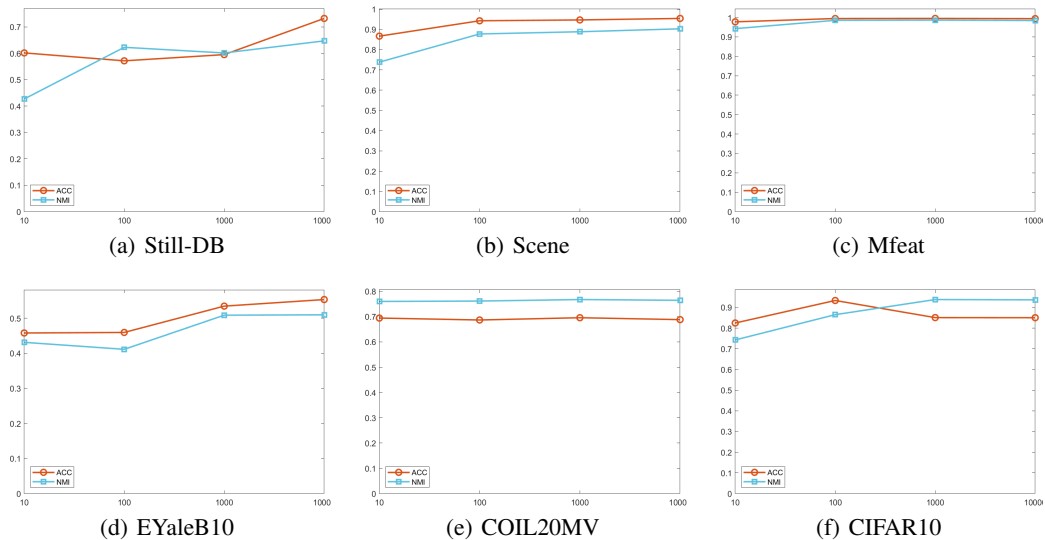

Figure 9: Effect of parameter $\beta$ on the CAMEL model on the rest datasets.

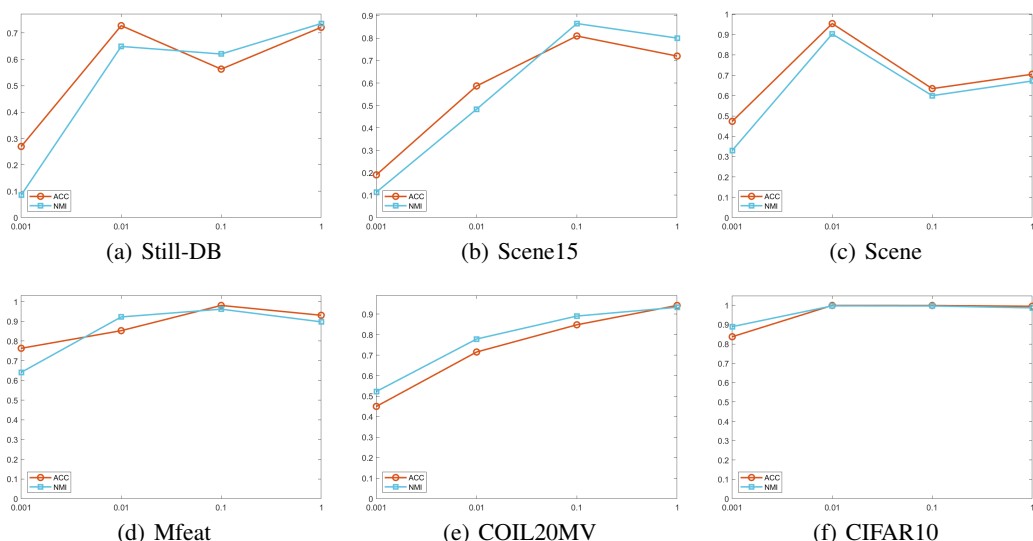

Figure 10: Effect of parameter $\delta$ on the CAMEL model on the rest datasets.

Table 5: Evaluation of the CAMEL model ablation on Still-DB, COIL20MV and Mfeat datasets.

| Components | | Still-DB | | | COIL20MV | | | Mfeat | | |
|---|---|---|---|---|---|---|---|---|---|---|
| CAMEL-C | CAMEL-T | ACC | NMI | PUR | ACC | NMI | PUR | ACC | NMI | PUR |
| ✓ | | 27.15 | 8.20 | 29.55 | 43.32 | 50.35 | 45.83 | 52.96 | 46.90 | 54.36 |
| | ✓ | 28.52 | 8.69 | 32.03 | 44.49 | 54.80 | 46.85 | 44.50 | 40.75 | 46.05 |
| ✓ | ✓ | **74.78** | **66.61** | **74.78** | **70.32** | **77.31** | **72.41** | **96.35** | **96.79** | **97.29** |

Table 6: Evaluation of the CAMEL model ablation on Scene, Scene15 and CIFAR10 datasets.

| Components | | Scene | | | Scene15 | | | CIFAR10 | | |
|---|---|---|---|---|---|---|---|---|---|---|
| CAMEL-C | CAMEL-T | ACC | NMI | PUR | ACC | NMI | PUR | ACC | NMI | PUR |
| ✓ | | 31.67 | 19.34 | 32.92 | 20.31 | 13.64 | 22.02 | 82.54 | 75.57 | 83.47 |
| | ✓ | 42.05 | 31.35 | 43.97 | 21.82 | 21.41 | 25.65 | 85.65 | 92.74 | 89.54 |
| ✓ | ✓ | **95.42** | **90.31** | **95.42** | **57.04** | **47.33** | **57.94** | **93.96** | **97.20** | **95.89** |

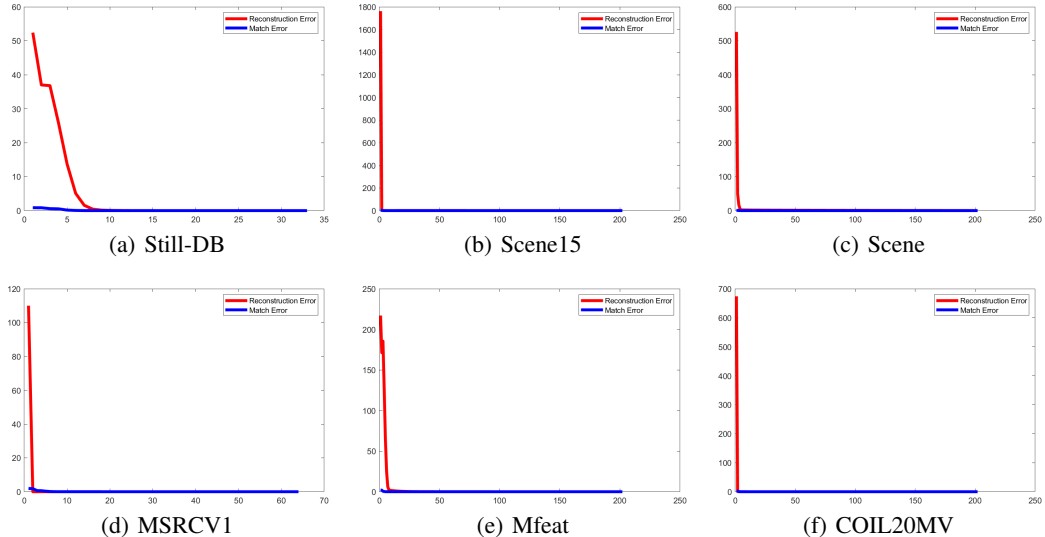

Figure 11: Training convergence trends of the CAMEL model on the rest datasets.

## A.5 THE IMPACT OF DIFFERENT NORMS ON CAMEL

We adopt the $\ell_\delta$-norm Kang et al. (2015) to address the theoretical limitations of the traditional tensor nuclear norm (TNN) Xie et al. (2018). As a convex relaxation, TNN imposes uniform penalties on all singular values, which tends to overly suppress principal information components and proves inadequate for noise removal. In contrast, the non-convex $\ell_\delta$-norm employed in this work offers adaptive penalization: it imposes stronger penalties on smaller singular values and weaker penalties on larger ones, thereby filtering more noise during low-rank approximation while better preserving essential information. We also acknowledge that numerous studies have proposed various other types of non-convex surrogates (e.g., Geman Geman & Yang (1995), Laplace Chen et al. (2021), Schatten $\ell_p$ Xia et al. (2022b), etc.). Fundamentally, these non-convex surrogates follow a similar underlying principle—assigning differentiated penalties to singular values to achieve a more refined characterization of the low-rank structure. Our choice of the $\ell_\delta$-norm is motivated by its empirically verified performance: across multiple datasets, it consistently delivers competitive results.

To further assess the effectiveness of the $\ell_\delta$-norm, we replaced it in our model with TNN and several representative non-convex surrogates (including Geman, Laplace, and Schatten $\ell_p$), and carefully tuned the parameters for each alternative. The comparative results are summarized in the table below(the missing rate is 0.5).

| Datasets | Yale3 | MSRCV1 | Still-DB | EYaleB10 | COIL20MV | Mfeat | Scene | Scene15 | CIFAR10 |
|---|---|---|---|---|---|---|---|---|---|
| TNN | 20.61±1.29 | 21.62±0.93 | 21.80±1.42 | 15.66±0.40 | 10.36±0.48 | 12.83±0.15 | 14.40±0.19 | 9.82±0.26 | 33.23±1.90 |
| Geman | 58.30±2.82 | 96.29±1.09 | 72.55±0.96 | 53.59±1.37 | 67.25±2.25 | 99.20±0.00 | 94.50±0.00 | 55.01±1.16 | 94.92±7.07 |
| Lapace | 33.21±3.87 | 41.33±3.06 | 29.42±1.17 | 26.56±1.28 | 46.28±3.99 | 60.51±4.29 | 33.90±0.83 | 18.23±1.16 | 90.13±2.19 |
| Schatten $\ell_p$ | 75.76±6.64 | 93.99±1.65 | 87.11±4.95 | 57.09±5.96 | 81.78±12.71 | 98.61±0.13 | 70.43±0.00 | 76.23±4.10 | 33.44±1.19 |
| $\ell_\delta$ | 61.82±2.27 | 97.90±0.26 | 74.78±0.46 | 55.72±5.42 | 70.32±1.33 | 96.35±6.62 | 95.42±0.00 | 57.04±3.30 | 93.96±8.10 |

Table 7: Experimental results on different norms.

The experimental results clearly show that, compared with the traditional TNN, employing any non-convex surrogate within the CAMEL framework leads to improved clustering performance. This demonstrates that non-convex penalties, by adaptively treating singular values, effectively preserve informative components while suppressing noise. Moreover, although the performance of different non-convex functions varies across datasets, the proposed $\ell_\delta$-norm consistently exhibits strong competitiveness: it achieves the best performance on MSRCV1 and Scene, and ranks second on datasets such as Yale3, Still-DB, EYaleB10, COIL20MV, Scene15, and CIFAR10. These findings confirm that adopting the $\ell_\delta$-norm for tensor low-rank modeling is both reasonable and effective. We would also like to clarify that the primary contribution of our work does not lie in designing

a particular non-convex surrogate. Rather, the main novelty is our constant-degree–driven connectivity constraint integrated with a low-rank constraint on the latent embedding tensor. The tensor rank surrogate itself is modular within our framework, allowing different functions to be seamlessly incorporated without altering the overall architecture. The experimental evidence demonstrates the extensibility and flexibility of our method: it accommodates various tensor low-rank surrogates while maintaining linear computational complexity, consistently delivering robust and competitive performance.

## A.6 THE ASPECTS IN WHICH **H** PROVIDES ADVANTAGES OVER **Z**

First, in terms of representation quality, each row of $\mathbf{H} \in \mathbb{R}^{n \times c}$ corresponds to a sample's embedding in a $c$-dimensional latent space, where $c$ denotes the number of clusters. Each dimension directly reflects the association strength of the sample to a specific cluster, providing clear semantic interpretability, and can thus be regarded as an effective cluster indicator matrix. In contrast, $\mathbf{Z} \in \mathbb{R}^{t \times n}$ represents the coordinates of samples in a subspace spanned by $t$ anchor points. Since $t$ is typically much larger than $c$, $\mathbf{Z}$ lacks direct characterization of cluster structure and exhibits semantic ambiguity. Second, in terms of clustering performance, since $\mathbf{H}$ inherently contains cluster-level structural information, applying $k$-means directly to it yields high-quality clustering results. This eliminates the need for post-processing steps(e.g., SVD) required by methods based on $\mathbf{Z}$ to extract implicit structures, thereby avoiding associated errors and computational overhead, and enhancing the method's efficiency and stability.

To further validate the advantage of $\mathbf{H}$ over $\mathbf{Z}$ in the CAMEL framework, we conducted additional clustering experiments using $\mathbf{Z}$ (referred to as CAMEL-Z) and compared them with the original CAMEL model. As shown in Table 8 below, CAMEL-Z consistently underperforms CAMEL across all datasets, further demonstrating the superiority of using $\mathbf{H}$.

Table 8: Comparison of clustering results of CAMEL-Z and CAMEL across varying levels of missing data.

| Data | Methods | 0.1 | | | 0.3 | | | 0.5 | | |
|---|---|---|---|---|---|---|---|---|---|---|
| | | ACC | NMI | PUR | ACC | NMI | PUR | ACC | NMI | PUR |
| Yale3 | CAMEL-Z | 52.73±1.77 | 53.96±1.29 | 53.21±0.79 | 48.00±1.84 | 51.55±2.59 | 48.97±1.17 | 34.67±5.85 | 37.72±5.36 | 36.85±5.11 |
| | CAMEL | 74.79±4.90 | 79.25±3.19 | 75.64±3.41 | 72.61±5.02 | 76.21±4.15 | 73.21±5.13 | 61.82±2.27 | 65.17±2.53 | 63.03±2.54 |
| MSRCV1 | CAMEL-Z | 77.81±4.19 | 69.52±2.23 | 78.67±3.15 | 60.67±5.11 | 51.41±2.24 | 63.43±3.82 | 40.76±4.28 | 31.18±2.83 | 44.29±3.64 |
| | CAMEL | 97.81±0.43 | 95.61±0.58 | 97.81±0.43 | 97.14±0.95 | 94.05±1.39 | 97.14±0.95 | 97.90±0.26 | 96.03±0.59 | 97.90±0.26 |
| Still-DB | CAMEL-Z | 30.62±0.96 | 11.06±0.92 | 31.99±1.22 | 32.33±1.12 | 10.80±0.30 | 34.99±1.03 | 30.02±0.55 | 10.41±0.76 | 30.75±0.72 |
| | CAMEL | 59.36±1.88 | 45.34±1.41 | 61.88±1.60 | 67.45±0.30 | 53.47±0.39 | 67.45±0.30 | 74.78±0.46 | 66.61±0.40 | 74.78±0.46 |
| EYaleB10 | CAMEL-Z | 22.75±1.49 | 13.89±1.86 | 24.00±1.04 | 21.44±1.61 | 12.38±1.78 | 22.44±1.60 | 18.03±1.75 | 6.63±2.04 | 19.28±1.63 |
| | CAMEL | 51.97±1.78 | 51.09±0.98 | 52.44±1.58 | 48.59±4.73 | 47.11±4.73 | 48.94±4.49 | 55.72±5.42 | 51.21±5.48 | 55.78±5.35 |
| COIL20MV | CAMEL-Z | 63.31±3.66 | 71.77±2.21 | 65.14±3.39 | 48.57±2.44 | 57.76±2.60 | 51.26±2.20 | 39.83±0.97 | 49.57±0.95 | 42.19±0.53 |
| | CAMEL | 68.38±2.27 | 79.17±0.98 | 71.22±1.84 | 69.28±1.77 | 77.67±0.98 | 70.81±1.52 | 70.32±1.33 | 77.31±0.70 | 72.41±0.95 |
| Mfeat | CAMEL-Z | 73.20±1.78 | 65.65±1.03 | 73.42±1.58 | 53.88±1.69 | 48.78±1.29 | 55.58±1.54 | 37.55±1.51 | 33.97±1.94 | 39.33±1.84 |
| | CAMEL | 99.65±0.00 | 99.04±0.00 | 99.65±0.00 | 96.97±6.16 | 97.99±2.81 | 97.72±4.48 | 96.35±6.62 | 96.79±6.62 | 97.29±4.52 |
| Scene | CAMEL-Z | 49.35±0.31 | 36.76±0.22 | 51.09±0.58 | 43.33±0.00 | 29.35±0.17 | 44.44±0.12 | 31.53±0.57 | 19.36±0.62 | 32.87±0.59 |
| | CAMEL | 97.29±0.00 | 93.54±0.00 | 97.29±0.00 | 95.99±0.00 | 91.64±0.00 | 95.99±0.00 | 95.42±0.00 | 90.31±0.13 | 95.42±0.00 |
| Scene15 | CAMEL-Z | 28.15±0.61 | 26.38±0.26 | 31.34±0.61 | 23.30±0.80 | 22.74±0.88 | 26.57±0.84 | 21.02±0.36 | 20.38±0.48 | 24.58±0.58 |
| | CAMEL | 72.23±2.02 | 69.10±0.99 | 75.21±1.81 | 61.17±0.97 | 54.00±1.63 | 62.51±1.85 | 57.04±3.30 | 47.33±1.13 | 57.94±1.67 |
| CIFAR10 | CAMEL-Z | 98.59±0.00 | 96.23±0.00 | 98.59±0.00 | 80.10±6.31 | 76.46±2.73 | 80.10±6.31 | 65.19±3.97 | 64.99±2.63 | 68.60±2.85 |
| | CAMEL | 99.99±0.00 | 99.95±0.00 | 99.99±0.00 | 99.95±0.00 | 99.82±0.00 | 99.95±0.00 | 93.96±8.10 | 97.20±3.26 | 95.89±5.46 |

