# OpenReview forum: "Constant Degree Matrix-Driven Incomplete Multi-View Clustering via Connectivity-Structure and Embedding Tensor Learning"
_ICLR.cc/2026/Conference — ICLR 2026 Poster_

### Official Review · Reviewer_Q7LQ · 2025-10-29

**Soundness:** 3
**Presentation:** 3
**Contribution:** 3
**Rating:** 6
**Confidence:** 4

**Summary:**

To effectively model the rich information in multi-view data while improving computational efficiency, this paper develops a constant-degree matrix-based framework for incomplete multi-view clustering. View-specific latent embeddings are learned under structured constraints and organized into a tensor with an $\ell_\delta$ low-rank constraint, enabling joint optimization of graph connectivity and high-order correlations. CAMEL further reduces the $O(n^2)$ complexity of traditional connectivity constraints by approximating the variable Laplacian degree matrix with a constant-degree matrix. Theoretical analysis and extensive experiments validate the method’s effectiveness.

**Strengths:**

1. The paper is well-written and presents the work in a clear and rigorous manner.
2. By approximating the variable Laplacian degree matrix with a constant-degree matrix, the proposed method avoids the $O(n^2)$ or higher computational complexity associated with traditional connectivity constraints, thereby improving overall efficiency. The authors also validate the approach on large-scale datasets, demonstrating its practical applicability.
3. The CAMEL consistently achieves superior clustering performance compared to baseline methods across multi-view datasets of different types.

**Weaknesses:**

1. One of the core contributions of this work is the use of the latent representation H instead of the subspace representation Z for clustering. It would be helpful if the authors could clarify in which aspects H provides advantages over Z, such as representation quality, or clustering performance.
2. The paper replaces the traditional Laplacian matrix $L_v = I - (D_v)^{-1/2} S_v (D_v)^{-1/2}$ with a constant form $L_v = I - \beta (Z_v)^\top Z_v$, but the underlying mechanism and justification for this approximation are insufficiently discussed.
3. The parameter sensitivity analysis uses different ranges for $\lambda_1$ and $\lambda_2$, but the rationale for selecting these ranges is not clearly justified.
4. The construction of incomplete multi-view data is insufficiently described. It is unclear whether the missing elements are consistent across different datasets and compared methods, and whether the randomly removed elements are the same across experimental runs.
5. The paper refers to the dataset as "Still2", while other references consistently use "Still-DB". Could the authors clarify whether these names refer to the same dataset or different ones?

**Questions:**

Please refer to the weaknesses mentioned above.

---

> ### Author Response · Authors · 2025-11-25
>
> We sincerely thank the reviewer for their valuable feedback. Our detailed responses to the points raised are provided below:
>
> >**Weakness 1: One of the core contributions of this work is the use of the latent representation H instead of the subspace representation Z for clustering. It would be helpful if the authors could clarify in which aspects H provides advantages over Z, such as representation quality, or clustering performance.**
>
> **A1**: Thank you for your insightful comment.  First, in terms of representation quality, each row of $\mathbf{H} \in \mathbb{R}^{n \times c}$ corresponds to a sample’s embedding in a $c$-dimensional latent space, where $c$ denotes the number of clusters. Each dimension directly reflects the association strength of the sample to a specific cluster, providing clear semantic interpretability, and can thus be regarded as an effective cluster indicator matrix. In contrast, $\mathbf{Z} \in \mathbb{R}^{t \times n}$ represents the coordinates of samples in a subspace spanned by $t$ anchor points. Since $t$ is typically much larger than $c$, $\mathbf{Z}$ lacks direct characterization of cluster structure and exhibits semantic ambiguity.
>
> Second, in terms of clustering performance, since $\mathbf{H}$ inherently contains cluster-level structural information, applying $k$-means directly to it yields high-quality clustering results. This eliminates the need for post-processing steps(e.g., SVD) required by methods based on $\mathbf{Z}$ to extract implicit structures, thereby avoiding associated errors and computational overhead, and enhancing the method's efficiency and stability.
>
> To further validate the advantage of $\mathbf{H}$ over $\mathbf{Z}$ in the CAMEL framework, we conducted additional clustering experiments using $\mathbf{Z}$ (referred to as CAMEL-Z) and compared them with the original CAMEL model. As shown in the tables below, CAMEL-Z consistently underperforms CAMEL across all datasets, further demonstrating the superiority of using $\mathbf{H}$.
>
> **Yale3**
> |Missing Rate  | 0.1 | | | 0.3 | | | 0.5 | | |
> |-----------------------|-----|-|-|-----|-|-|-----|-|-|
> | **Methods**      | ACC | NMI | PUR | ACC | NMI | PUR | ACC | NMI | PUR |
> | CAMEL-Z           | 52.73±1.77 | 53.96±1.29 | 53.21±0.79 | 48.00±1.84 | 51.55±2.59 | 48.97±1.17 | 34.67±5.85 | 37.72±5.36 | 36.85±5.11 |
> | CAMEL           | 74.79±4.90 | 79.25±3.19 | 75.64±3.41 | 72.61±5.02 | 76.21±4.15 | 73.21±5.13 | 61.82±2.27 | 65.17±2.53 | 63.03±2.54 |
>
> **MSRCV1**
> |Missing Rate  | 0.1 | | | 0.3 | | | 0.5 | | |
> |-----------------------|-----|-|-|-----|-|-|-----|-|-|
> | **Methods**      | ACC | NMI | PUR | ACC | NMI | PUR | ACC | NMI | PUR |
> | CAMEL-Z           | 77.81±4.19 | 69.52±2.23 | 78.67±3.15 | 60.67±5.11 | 51.41±2.24 | 63.43±3.82 | 40.76±4.28 | 31.18±2.83 | 44.29±3.64 |
> | CAMEL           | 97.81±0.43 | 95.61±0.58 | 97.81±0.43 | 97.14±0.95 | 94.05±1.39 | 97.14±0.95 | 97.90±0.26 | 96.03±0.59 | 97.90±0.26 |
>
> **Still-2**
> |Missing Rate  | 0.1 | | | 0.3 | | | 0.5 | | |
> |-----------------------|-----|-|-|-----|-|-|-----|-|-|
> | **Methods**      | ACC | NMI | PUR | ACC | NMI | PUR | ACC | NMI | PUR |
> | CAMEL-Z           | 30.62±0.96 | 11.06±0.92 | 31.99±1.22 | 32.33±1.12 | 10.80±0.30 | 34.99±1.03 | 30.02±0.55 | 10.41±0.76 | 30.75±0.72 |
> | CAMEL           | 59.36±1.88 | 45.34±1.41 | 61.88±1.60 | 67.45±0.30 | 53.47±0.39 | 67.45±0.30 | 74.78±0.46 | 66.61±0.40 | 74.78±0.46 |
>
> **EYaleB10**
> |Missing Rate  | 0.1 | | | 0.3 | | | 0.5 | | |
> |-----------------------|-----|-|-|-----|-|-|-----|-|-|
> | **Methods**      | ACC | NMI | PUR | ACC | NMI | PUR | ACC | NMI | PUR |
> | CAMEL-Z           | 22.75±1.49 | 13.89±1.86 | 24.00±1.04 | 21.44±1.61 | 12.38±1.78 | 22.44±1.60 | 18.03±1.75 | 6.63±2.04 | 19.28±1.63 |
> | CAMEL           | 51.97±1.78 | 51.09±0.98 | 52.44±1.58 | 48.59±4.73 | 47.11±4.73 | 48.94±4.49 | 55.72±5.42 | 51.21±5.48 | 55.78±5.35 |
>
> **COIL20MV**
> |Missing Rate  | 0.1 | | | 0.3 | | | 0.5 | | |
> |-----------------------|-----|-|-|-----|-|-|-----|-|-|
> | **Methods**      | ACC | NMI | PUR | ACC | NMI | PUR | ACC | NMI | PUR |
> | CAMEL-Z           | 63.31±3.66 | 71.77±2.21 | 65.14±3.39 | 48.57±2.44 | 57.76±2.60 | 51.26±2.20 | 39.83±0.97 | 49.57±0.95 | 42.19±0.53 |
> | CAMEL           | 68.38±2.27 | 79.17±0.98 | 71.22±1.84 | 69.28±1.77 | 77.67±0.98 | 70.81±1.52 | 70.32±1.33 | 77.31±0.70 | 72.41±0.95 |

---

> ### Author Response · Authors · 2025-11-25
>
> **Mfeat**
> |Missing Rate  | 0.1 | | | 0.3 | | | 0.5 | | |
> |-----------------------|-----|-|-|-----|-|-|-----|-|-|
> | **Methods**      | ACC | NMI | PUR | ACC | NMI | PUR | ACC | NMI | PUR |
> | CAMEL-Z           | 73.20±1.78 | 65.65±1.03 | 73.42±1.58 | 53.88±1.69 | 48.78±1.29 | 55.58±1.54 | 37.55±1.51 | 33.97±1.94 | 39.33±1.84 |
> | CAMEL           | 99.65±0.00 | 99.04±0.00 | 99.65±0.00 | 96.97±6.16 | 97.99±2.81 | 97.72±4.48 | 96.35±6.62 | 96.79±6.62 | 97.29±4.52 |
>
> **Scene**
> |Missing Rate  | 0.1 | | | 0.3 | | | 0.5 | | |
> |-----------------------|-----|-|-|-----|-|-|-----|-|-|
> | **Methods**      | ACC | NMI | PUR | ACC | NMI | PUR | ACC | NMI | PUR |
> | CAMEL-Z           | 49.35±0.31 | 36.76±0.22 | 51.09±0.58 | 43.33±0.00 | 29.35±0.17 | 44.44±0.12 | 31.53±0.57 | 19.36±0.62 | 32.87±0.59 |
> | CAMEL          | 97.29±0.00 | 93.54±0.00 | 97.29±0.00 | 95.99±0.00 | 91.64±0.00 | 95.99±0.00 | 95.42±0.00 | 90.31±0.13 | 95.42±0.00 |
>
> **Scene15**
> |Missing Rate  | 0.1 | | | 0.3 | | | 0.5 | | |
> |-----------------------|-----|-|-|-----|-|-|-----|-|-|
> | **Methods**      | ACC | NMI | PUR | ACC | NMI | PUR | ACC | NMI | PUR |
> | CAMEL-Z           | 28.15±0.61 | 26.38±0.25 | 31.34±0.61 | 23.30±0.80 | 22.74±0.88 | 26.57±0.84 | 21.02±0.36 | 20.38±0.48 | 24.58±0.58 |
> | CAMEL          | 72.23±2.02 | 69.10±0.99 | 75.21±1.81 | 61.17±0.97 | 54.00±1.63 | 62.51±1.85 | 57.04±3.30 | 47.33±1.13 | 57.94±1.67 |
>
> **CIFAR10**
> |Missing Rate  | 0.1 | | | 0.3 | | | 0.5 | | |
> |-----------------------|-----|-|-|-----|-|-|-----|-|-|
> | **Methods**      | ACC | NMI | PUR | ACC | NMI | PUR | ACC | NMI | PUR |
> | CAMEL          | 98.59±0.00 | 96.23±0.00 | 98.59±0.00 | 80.10±6.31 | 76.46±2.73 | 80.10±6.31 | 65.19±3.97 | 64.99±2.63 | 68.60±2.85 |
> | CAMEL           | 99.99±0.00 | 99.95±0.00 | 99.99±0.00 | 99.95±0.00 | 99.82±0.00 | 99.95±0.00 | 93.96±8.10 | 97.20±3.26 | 95.89±5.46 |
> ___
> >**Weakness 2: The paper replaces the traditional Laplacian matrix with a constant form, but the underlying mechanism and justification for this approximation are insufficiently discussed.**
>
> **A2**: We thank the reviewer for this insightful comment. In our model, the Laplacian matrix is computed based on the similarity matrix $\mathbf{Z}^v$, which inherently contains noise. Consequently, the corresponding degree matrix, whether learnable or constant, inevitably incorporates noise.  In certain scenarios, using a constant degree matrix as an approximation may not degrade performance. Importantly, adopting a constant degree matrix reduces computational complexity and avoids potential instability. Since actual noise levels are dataset-dependent and cannot be directly measured, the choice between a learnable or constant degree matrix should be guided by practical considerations.
> ___
> >**Weakness 3: The parameter sensitivity analysis uses different ranges for $\lambda_1$ and $\lambda_2$, but the rationale for selecting these ranges is not clearly justified.**
>
> **A3**: Thank you for your thoughtful comment. In our study, the parameter ranges were initially determined through empirical testing, which is a common practice in the field for establishing reasonable search intervals.
> ___
> >**Weakness 4: The construction of incomplete multi-view data is insufficiently described. It is unclear whether the missing elements are consistent across different datasets and compared methods, and whether the randomly removed elements are the same across experimental runs.**
>
> **A4**: Thank you for your detailed comment. All datasets are first processed into incomplete-view versions, and these incomplete-view datasets remain unchanged in subsequent experiments. For example, the original Yale3 dataset is first processed into an incomplete-view dataset Yale3_i, and all subsequent experiments are conducted on this fixed Yale3_i without further modifications.
> ___
> >**Weakness 5: The paper refers to the dataset as "Still2", while other references consistently use "Still-DB". Could the authors clarify whether these names refer to the same dataset or different ones?**
>
> **A5**: Thank you. The Still2 dataset in our paper corresponds to the Still-DB dataset used in other works. We will clarify this in the revised manuscript to avoid any ambiguity.

---

### Official Review · Reviewer_q5uJ · 2025-10-30

**Soundness:** 3
**Presentation:** 3
**Contribution:** 3
**Rating:** 6
**Confidence:** 5

**Summary:**

This paper proposes a tensor-based method for incomplete multi-view clustering named CAMEL. CAMEL jointly learns view-specific latent embeddings under structured constraints and organizes them into a low-rank tensor, enabling coordinated optimization of graph connectivity and high-order correlations while avoiding costly post-processing (e.g., SVD). Extensive experiments on nine benchmark datasets demonstrate its superior effectiveness and efficiency compared to existing methods.

**Strengths:**

1.The proposed method learns view-specific embeddings under structured constraints and organizes them into a low-rank tensor, effectively overcoming the limitation of traditional methods that cannot simultaneously capture graph connectivity and high-order correlations.

2.The manuscript presents extensive experiments that demonstrate the effectiveness and efficiency of the proposed method.

3.The manuscript provides detailed theoretical derivations and proofs, which theoretically validate the effectiveness of the proposed model.

**Weaknesses:**

1. The hyperparameters $\lambda_1$ and $\lambda_2$ in Equations (1)--(3) are not defined, leaving their values and roles unclear.

2. The manuscript lacks experimental comparisons between the $\ell_\delta$-norm and the conventional tensor nuclear norm in terms of performance.

3. The authors state that Equation (2) involves quadratic computational complexity; however, since Equation (2) is based on an anchor method, it should already have low complexity. It is unclear where the quadratic complexity arises.

4. Lines 259--262 contain a repeated statement, which affects the manuscript’s clarity.

**Questions:**

See Weakness section.

---

> ### Author Response · Authors · 2025-11-25
>
> We sincerely appreciate your thoughtful and constructive comments. We address each of the points raised below.
>
> >**Weakness 1: The hyperparameters $\lambda_1$ and $\lambda_2$ in Equations (1)--(3) are not defined, leaving their values and roles unclear.**
>
> **A1**: Thank you for your careful observations. The two hyperparameters serve solely for trade-off purposes, with their values specified in the experimental section: $\lambda_1 \in [10^{-8},10^{-3}]$ and $\lambda_2 \in [10^{-4},10^{-2}]$.
>
> >**Weakness 2: The manuscript lacks experimental comparisons between the  $\ell_{\delta}$-norm and the conventional tensor nuclear norm in terms of performance.**
>
> **A2**: Thank you for your constructive comment. We replaced the $\ell_{\delta}$-norm in our model with the conventional tensor nuclear norm (TNN) and fine-tuned the parameters to achieve the best clustering performance under this setting. We report the average ACC over 5 runs on datasets with a 50% missing rate.  Experiments were conducted on nine datasets, and the results are shown in the table below.
>
> | Datasets | Yale3 | MSRC_v1 | Still-2 | EYaleB10 | COIL20MV | Mfeat | Scene | Scene15 | CIFAR10 |
> |------------|---------|---------------|---------|---------------|-----------------|---------|----------|-------------|-------------|
> | TNN |  20.61±1.29  | 21.62±0.93 | 21.80±1.42 | 15.66±0.40 | 10.36±0.48 | 12.83±0.15 | 14.40±0.19 | 9.82±0.26 | 33.23±1.90
> | $\ell_\delta$ | 61.82±2.27 | 97.90±0.26 | 74.78±0.46 | 55.72±5.42 | 70.32±1.33 | 96.35±6.62 | 95.42±0.00 | 57.04±3.30 | 93.96± 8.10
>
> It can be observed that replacing the $\ell_{\delta}$-norm with TNN leads to a substantial drop in clustering performance across all nine datasets, indicating that, compared to TNN, the $\ell_{\delta}$-norm imposes stronger penalties on smaller singular values while reducing penalties on larger ones, thereby filtering more noise during low-rank approximation and better preserving crucial information. Furthermore, this demonstrates that our CAMEL model not only integrates the constant-degree-driven connectivity constraint with a low-rank constraint on the latent embedding tensor to enhance performance and efficiency, but also can seamlessly work with different tensor norms, highlighting its strong flexibility.
>
> >**Weakness 3: The authors state that Equation (2) involves quadratic computational complexity; however, since Equation (2) is based on an anchor method, it should already have low complexity. It is unclear where the quadratic complexity arises.**
>
> **A3**:In our model, the anchor-based approach itself does not possess quadratic complexity. As indicated in Section 4.3, the computational complexity of calculating $A^v$ is $\mathcal{O}(d_vtn + d_vt)$. The quadratic complexity arises during the computation of the Laplacian matrix $\mathbf{L}^v$. Specifically, when computing $\mathbf{L}^v$, its degree matrix $\mathbf{D}^v$ must be derived. The calculation of $\mathbf{D}^v$ requires column-wise summation of $\mathbf{S}^v \in \mathbb{R}^{n \times n}$, which scales as $\mathcal{O}(n^2)$. This quadratic complexity arises because each of the n columns in $\mathbf{S}^v$ contains n elements that need to be summed, resulting in n×n operations.
> ___
> >**Weakness 4: Lines 259--262 contain a repeated statement, which affects the manuscript’s clarity.**
>
> **A4**:We thank the reviewer for this careful observation. We acknowledge the repetition in Lines 259–262 and will remove the redundant statement in the final version to improve the manuscript's clarity and conciseness.

---

### Official Review · Reviewer_MEf5 · 2025-10-30

**Soundness:** 2
**Presentation:** 2
**Contribution:** 1
**Rating:** 2
**Confidence:** 5

**Summary:**

This paper proposes a novel incomplete multi-view clustering method that seamlessly integrates the low-rank tensor learning, latent learning, anchor learning as one unified framework. However, this paper is with limited contributions, since the proposed method is the combination of existing methods such as "Tensorized multi-view subspace representation learning", "Generalized latent multi-view subspace clustering", and so on.

**Strengths:**

The paper is well-written.
The authors also illustrate the motivations, details, and optimization of the proposed method.

**Weaknesses:**

1、In Figure 1, what is the function of “Constant degree matrix”? How to communicate with other modules of the proposed method?
2、Many incomplete and complete multi-view clustering methods assume that there exists some sparse noise such as deadline. The authors select the ||.||_F as the regularizer for noise removal. How to deal with the datasets with sparse noise?
3、How to extend the l_\sigma norm from the matrix format into the tensor format? How to define the l_\sigma tensor norm? There are many nonconvex surrogates such as l_p norm, Schatten l_p norm, and so on, for nonconvex low-rank tensor learning. The authors failed to compare the proposed method with other representative nonconvex surrogates for high-order modeling.
 4、Small-scale testing datasets. The testing datasets are relatively small-scale! I want to see the performance of the proposed method on the large-scale dataset.

**Questions:**

In Equation ( 3), the authors have added the "Tr[(Hv )⊤(I − 1β (Zv )⊤Zv )Hv" into the objective of the proposed model. The author states that the proposed model could reduce the running times. However, I think the introduction of  "Tr[(Hv )⊤(I − 1β (Zv )⊤Zv )Hv"  could yield a heavy computational burden, rather than reduce the running times.

---

> ### Author Response · Authors · 2025-11-20
>
> We sincerely thank you for your thorough and thoughtful review of our manuscript and for providing valuable comments that have helped us improve the work. Below, we provide point-by-point responses to the concerns you raised.
>
> >**Weakness 1: In Figure 1, what is the function of “Constant degree matrix”? How to communicate with other modules of the proposed method?**
>
> **A1**: We thank the reviewer for the insightful questions regarding the function and integration of the "constant degree matrix".① The "Constant degree matrix" functions as a computational shortcut, replacing the complex and time-consuming calculation of the degree matrix $\mathbf{D}^v$ (which involves column-wise summation of the affinity matrix $\mathbf{S}^v$) with a simple, fixed matrix $\beta \mathbf{I}$. This substitution significantly reduces the $\mathcal{O}(n^2)$ computational bottleneck, improving the algorithm’s efficiency, particularly for large datasets.
>
> ②In our method,  the "Constant degree matrix" is utilized to compute the Laplacian matrix for each graph. The Laplacian matrix is calculated according to $\mathbf{L}^v = \mathbf{I} - (\mathbf{D}^v)^{-\frac{1}{2}} \mathbf{S}^v (\mathbf{D}^v)^{-\frac{1}{2}}$, where $\mathbf{S}^v = (\mathbf{Z}^v)^{\top} \mathbf{Z}^v$. For all views, the degree matrix $\mathbf{D}^v$ is substituted with a constant degree matrix $\mathbf{D}$, resulting in $\mathbf{D}^v = \mathbf{D} = \beta \mathbf{I}$, hence the Laplacian matrix calculation simplifies to $\mathbf{L}^v = \mathbf{I} - \tfrac{1}{\beta}(\mathbf{Z}^v)^{\top}\mathbf{Z}^v$. This demonstrates the involvement of the constant degree matrix.

---

> > ### Author Response · Authors · 2025-11-20
> >
> > >**Weakness 2: Many incomplete and complete multi-view clustering methods assume that there exists some sparse noise such as deadline. The authors select the ||.||F as the regularizer for noise removal. How to deal with the datasets with sparse noise?**
> >
> > **A2**: Thank you for this helpful comment. We would like to clarify that the term “deadline’’ in the comment might be a typo. We assume the reviewer may be referring to “dead pixel’’ or other forms of sparse corruption, which are commonly modeled through sparsity-inducing norms. Sparse noise generally involves only a small number of corrupted data points, while the majority of the data remains intact. However, in our setting of incomplete multi-view clustering (IMVC), the primary challenge stems from the incompleteness of the dataset, which results in a substantial portion of the data being missing. In this case, the key issue to address is the noise introduced by missing data rather than sparse noise. Therefore, when only one noise norm can be selected, we use the Frobenius norm to mitigate the negative impact of missing data. Other methods also employ the Frobenius norm for the same purpose, such as "Incomplete multi-modal visual data grouping[1]" and "Multiple incomplete views clustering via weighted nonnegative matrix factorization with L2,1 regularization[2]."
> >
> > >**Weakness 3: How to extend the l\sigma norm from the matrix format into the tensor format? How to define the l_\sigma tensor norm? There are many nonconvex surrogates such as l_p norm, Schatten l_p norm, and so on, for nonconvex low-rank tensor learning. The authors failed to compare the proposed method with other representative nonconvex surrogates for high-order modeling.**
> >
> > **A3**:We appreciate the reviewer's constructive comment. ①② We believe there may have been a misunderstanding with the notation: the norm used in our method is the $\ell_{\delta}$ norm, not the $\ell_{\sigma}$ norm as suggested.
> > The $\ell_{\delta}$ norm was introduced in "Robust PCA via Nonconvex Rank Approximation[3]" and we employ it as a rank function (i.e., $f(x) = \tfrac{(1+\delta)x}{\delta+x}$).  The $\ell_{\delta}$ norm in matrix format is commonly used in low-rank approximation, where it serves as a surrogate for the rank function. For extending this to tensors, the basic idea is to generalize the notion of the "singular values" of a matrix to the "tensor singular values" of higher-order tensors, which can then be regularized using a similar $\ell_{\delta}$-based rank function.
> >
> > For a third-order tensor $\boldsymbol{\mathcal{H}} \in \mathbb{R}^{n_1 \times n_2 \times n_3}$, its $\ell_{\delta}$-norm is mathematically expressed as follows:
> > $$
> > \|\| \boldsymbol{\mathcal{H}}\|\|\_\mathrm{\ell\_{\delta}} = \frac{1}{n_3} \sum_{k=1}^{n_3} \|\|\boldsymbol{\mathcal{H}}\_f^k\|\|\_\mathrm{\ell\_{\delta}} = \frac{1}{n_3} \sum_{k=1}^{n_3} \sum_{i=1}^{h} \frac{(1+\delta)\boldsymbol{\mathcal{S}}_f^k(i,i)}{\delta + \boldsymbol{\mathcal{S}}_f^k(i,i)}
> > $$
> > In this formulation, $h$ is defined as the minimum of $n_1$ and $n_2$, while the positive scalar $\delta$ serves as a parameter of the $\ell\_{\delta}$ rank function. The matrix $\boldsymbol{\mathcal{S}}_f^k$ corresponds to the $k$-th frontal slice of the tensor $\boldsymbol{\mathcal{S}}_f$, derived from the tensor singular value decomposition (t-SVD) applied to the $k$-th frontal slice of $\boldsymbol{\mathcal{H}}$, following the model $\boldsymbol{\mathcal{H}}_f^k = \boldsymbol{\mathcal{U}}_f^k \boldsymbol{\mathcal{S}}_f^k (\boldsymbol{\mathcal{V}}_f^k)^\top$.  This generalization allows us to apply the matrix norm to the entire tensor, where each frontal slice is treated like a matrix, and the norm is computed over all the slices.

---

> ### Author Response · Authors · 2025-11-20
>
> ③ Regarding the comparison of different tensor nonconvex surrogates for high-order modeling, we would like to clarify that the contribution of our work lies not in designing or selecting a specific nonconvex surrogate, but in proposing a framework driven by a constant degree matrix that integrates connectivity constraints with tensor representation learning. This framework enables the model to simultaneously capture the geometric structure of each view, model high-order correlations across views, and maintain high computational efficiency. In contrast, simply adopting a particular nonconvex or convex surrogate is neither the focus nor the core contribution of this work. However, in response to the reviewer's suggestion, we have conducted additional experiments using other representative nonconvex surrogates within our model, including: TNN [4], Geman [5], Laplace [6], and Schatten $\ell_p$ [7]. We tuned the parameters for each surrogate and report the best clustering performance with a 50% missing rate. The experiments were repeated five times, and the average ACC is as follows:
>
> | Datasets | Yale3 | MSRC_v1 | Still-2 | EYaleB10 | COIL20MV | Mfeat | Scene | Scene15 | CIFAR10 |
> |------------|---------|---------------|---------|---------------|-----------------|---------|----------|-------------|-------------|
> | TNN[4] |  20.61±1.29  | 21.62±0.93 | 21.80±1.42 | 15.66±0.40 | 10.36±0.48 | 12.83±0.15 | 14.40±0.19 | 9.82±0.26 | 33.23±1.90
> | Geman[5]| 58.30±2.82 | 96.29±1.09 | 72.55±0.96 | 53.59±1.37 | 67.25±2.25 | 99.20±0.00 | 94.50±0.00 | 55.01±1.16 | 94.92±7.07
> | Lapace[6] | 33.21±3.87 | 41.33±3.06 | 29.42±1.17 | 26.56±1.28 | 46.28±3.99 | 60.51±4.29 | 33.90±0.83 | 18.23±1.16 | 90.13±2.19
> | Schatten$\ell\_p$[7] | 75.76±6.64 | 93.99±1.65 | 87.11±4.95 | 57.09±5.96 | 81.78±12.71 | 98.61±0.13 | 70.43±0.00 | 76.23±4.10 | 33.44±1.19
> | $\ell \_δ$[3] | 61.82±2.27 | 97.90±0.26 | 74.78±0.46 | 55.72±5.42 | 70.32±1.33 | 96.35±6.62 | 95.42±0.00 | 57.04±3.30 | 93.96± 8.10
>
> The experimental results show that, although the innovation in terms of tensor norms is not the primary contribution of this work, the $\ell_{\delta}$ norm we adopt still demonstrates highly competitive performance. Specifically, on the MSRC_v1 and Scene datasets, the $\ell_{\delta}$ norm achieves the best clustering performance, while on Yale3, Still-2, EYaleB10, COIL20MV, Scene15, and CIFAR10, it ranks second. These results indicate that choosing the $\ell_{\delta}$ norm as the tensor norm is reasonable and further validates the effectiveness of our model. Moreover, the experiments highlight the excellent scalability of our method: it can accommodate different tensor norms while maintaining linear computational complexity, consistently achieving strong performance across all cases. This further demonstrates the flexibility and robustness of our approach.

---

> ### Author Response · Authors · 2025-11-20
>
> >**Weakness 4: Small-scale testing datasets. The testing datasets are relatively small-scale! I want to see the performance of the proposed method on the large-scale dataset.**
>
> **A4**: Thank you for your insightful comment.  In our original paper, we have already evaluated our method on the large-scale CIFAR10 dataset, which contains 50,000 samples and is considered quite large in the multi-view clustering community. The results on CIFAR10 demonstrate that our method achieves superior clustering performance on large-scale datasets. To further strengthen the evidence, we conducted additional experiments on the Noisy MNIST test set. Noisy MNIST consists of 70,000 samples, each with two views: the first view contains the original clean images, and the second view is generated by randomly selecting images from the same class and adding Gaussian noise. We used the standard 10,000-image test set for performance evaluation. All experiments were repeated five times with carefully tuned hyperparameters, and the averaged results are reported.
> |Missing Rate  | 0.1 | | | 0.3 | | | 0.5 | | |
> |-----------------------|-----|-|-|-----|-|-|-----|-|-|
> | **Methods**             | ACC | NMI | PUR | ACC | NMI | PUR | ACC | NMI | PUR |
> | BSV                   | 36.68±23.15 | 25.84±23.53 | 37.03±23.42 | 31.29±1.05 | 24.86±0.68 | 35.82±0.84 | 27.62±0.68 | 20.58±0.82 | 31.28±0.78 |
> | Concat                | 35.82±1.01 | 31.93±0.62 | 42.19±0.47 | 30.73±1.74 | 24.73±1.39 | 35.55±1.87 | 27.42±1.04 | 20.52±0.77 | 31.20±0.60 |
> | PVC                   | 39.96±2.99 | 33.93±4.23 | 43.39±3.38 | 31.86±1.98 | 23.21±0.87 | 34.65±1.16 | 35.63±4.24 | 25.68±4.12 | 38.27±3.84 |
> | IMVC-CBG         | 45.67±0.00 | 37.71±0.00 | 46.87±0.00 | 33.87±0.00 | 24.66±0.00 | 34.48±0.00 | 23.75±0.00 | 15.52±0.00 | 26.06±0.00 |
> | PSIMVC-PG       | 39.99±0.00 | 32.60±0.00 | 41.28±0.00 | 28.39±0.00 | 20.00±0.00 | 30.57±0.00 | 22.44±0.00 | 15.18±0.00 | 25.92±0.00 |
> | SCSL                  | 21.11±0.00 | 9.68±0.00 | 22.50±0.00 | 16.98±0.00 | 5.86±0.00 | 18.14±0.00 | 14.96±0.00 | 2.40±0.00 | 16.90±0.00 |
> | CAMEL               | 93.47±7.96 | 95.64±3.32 | 95.36±5.37 | 90.67±7.73 | 94.27±3.24 | 93.31±5.31 | 93.41±7.63 | 95.18±3.16 | 95.31±5.03 |
>
> As shown in the table, CAMEL significantly outperforms all baselines on Noisy MNIST across all missing rates. For example, under a missing rate of 0.1, CAMEL achieves an ACC that is 47.8% higher than the second-best model, IMVC-CBG, demonstrating its superior performance on large-scale datasets. Moreover, the computational complexity of our model scales linearly with the number of samples, indicating that our method not only achieves strong clustering performance on large-scale datasets but is also highly efficient, making it well-suited for large-scale applications.

---

> ### Author Response · Authors · 2025-11-25
>
> >**Question: In Equation ( 3), the authors have added the "Tr[(Hv )⊤(I − 1β (Zv )⊤Zv )Hv" into the objective of the proposed model. The author states that the proposed model could reduce the running times. However, I think the introduction of "Tr[(Hv )⊤(I − 1β (Zv )⊤Zv )Hv" could yield a heavy computational burden, rather than reduce the running times.**
>
> **A5**: We thank the reviewer for this question. The term $\mathrm{Tr}[(\mathbf{H}^v)^{\top}(\mathbf{I}-\frac{1}{\beta}(\mathbf{Z}^v)^{\top}\mathbf{Z}^v)\mathbf{H}^v]$ only relates to the two optimization variables $\mathbf{H}^v$ and $\mathbf{Z}^v$, and we only need to analyze their impact on complexity during the optimization process. Originally: For computing the optimizable Laplacian matrix, the corresponding degree matrix also needs to be optimized, the computation of the degree matrix $\mathbf{D}^v$ requires column-wise summation of $\mathbf{S}^v \in \mathbb{R}^{n \times n}$, which scales as $\mathcal{O}(n^2)$. After improvement: we replace the conventional variable degree matrix with a constant form, $\mathbf{D}=\beta \mathbf{I}$. The complexity of optimizing $\mathbf{H}^v$ in Algorithm 1 is $\mathcal{O}(tcn+c^2n)$, while the formula for optimizing $\mathbf{Z}^v$ is $\mathbf{Z}^v=\max(0,\frac{1}{\mu}\left[ (\mathbf{A}^v)^{\top}\mathbf{Y}^v+\mu(\mathbf{A}^v)^{\top}(\mathbf{X}^v-\mathbf{E}^v)\right])$, with a calculated time complexity of $\mathcal{O}(d_vn+d_vtn+tn)$. It can be seen that by using the constant degree matrix, the computational complexity of the Laplacian matrix no longer contains $\mathcal{O}(n^2)$, and analysis shows that the computational complexities of $\mathbf{H}^v$ and $\mathbf{Z}^v$ also maintain a linear relationship with the sample size. We have included detailed derivations in the "Detailed Optimization Process" subsection of the Appendix for the reviewer's reference, and we would be glad to provide any additional clarification if needed.
>
> References
>
> [1] H. Zhao, H. Liu, and Y. Fu, “Incomplete multi-modal visual data grouping,” in Proc. IJCAI, 2016, pp. 2392–2398.
>
> [2] W. Shao, L. He, and P. S. Yu, “Multiple incomplete views clustering via weighted nonnegative matrix factorization with L2,1 regularization,”in Proc. ECML PKDD, 2015, pp. 318–334.
>
> [3] Z. Kang, C. Peng, and Q. Cheng, “Robust PCA via nonconvex rank approximation,” in Proc. ICDM, 2015, pp. 211–220.
>
> [4] Y. Xie, D. Tao, W. Zhang, Y. Liu, L. Zhang, and Y. Qu, “On unifying multi-view self-representations for clustering by tensor multi-rank minimization,” International Journal of Computer Vision, vol. 126, no. 11, pp. 1157–1179, 2018.
>
>
> [5] D. Geman and C. Yang, “Nonlinear image recovery with half-quadratic regularization,” IEEE Transactions on Image Processing, vol. 4, no. 7, pp. 932–946, 1995.
>
> [6] Y. Chen, S. Wang, X. Xiao, Y. Liu, Z. Hua, and Y. Zhou, “Self-paced enhanced low-rank tensor kernelized multi-view subspace clustering,” IEEE Transactions on Multimedia, vol. 24, pp. 4054–4066, 2021.
>
> [7] W. Xia, Q. Gao, Q. Wang, X. Gao, C. Ding, and D. Tao, “Tensorized bipartite graph learning for multi-view clustering,” IEEE Transactions on Pattern Analysis and Machine Intelligence, vol. 45, pp. 5187–5202.

---

### Official Review · Reviewer_bSK1 · 2025-11-01

**Soundness:** 3
**Presentation:** 3
**Contribution:** 3
**Rating:** 6
**Confidence:** 5

**Summary:**

The manuscript develops a novel tensor-based approach for incomplete multi-view clustering tasks. The main ideas of the work are:
* Employ view-specific connectivity constraints and tensor learning to capture high-order cross-view correlations;
* Perform clustering on the learned embeddings to eliminate additional post-processing;
* Introduce a approximation of Laplacian matrix for efficiency.

**Strengths:**

* The work improves clustering performance and meanwhile enhances algorithm efficiency;
* Detailed theoretical analyses are provided, and experimental results are sufficient;
* The paper is well-written.

**Weaknesses:**

* Why the authors adopt the norm of $\ell_\delta$-norm (i.e., $f(x) = \frac{(1+\delta)x}{\delta + x}$) ? What its specific advangates?
* Some details require further explanation. For example, what is the clear definition of the "post-processing-free" methods? It seems that some post-processing-free methods still requires an extra clustering procedure on the learned features.
* The ablation study requires a deeper analyses, and current discussion cannot well illustrate the contribution of each component.
* Some typos should be corrected.

**Questions:**

Please refer to the weaknesses.

---

> ### Author Response · Authors · 2025-11-25
>
> We sincerely appreciate your constructive and insightful comments. Please find our detailed responses to your concerns below.
>
> **Weaknesses 1: Why the authors adopt the norm of $\ell_{\delta}$ norm?  What its specific advangates?**
>
> **A1**: Thank you for your constructive comment. We adopt the $\ell_{\delta}$-norm [1] to address the theoretical limitations of the traditional tensor nuclear norm (TNN) [2]. As a convex relaxation, TNN imposes uniform penalties on all singular values, which tends to overly suppress principal information components and proves inadequate for noise removal. In contrast, the non-convex $\ell_{\delta}$-norm employed in this work offers adaptive penalization: it imposes stronger penalties on smaller singular values and weaker penalties on larger ones, thereby filtering more noise during low-rank approximation while better preserving essential information. We also acknowledge that numerous studies have proposed various other types of non-convex surrogates (e.g., Geman[3], Laplace[4], Schatten $\ell_p$[5], etc.). Fundamentally, these non-convex surrogates follow a similar underlying principle—assigning differentiated penalties to singular values to achieve a more refined characterization of the low-rank structure. Our choice of the $\ell_{\delta}$-norm is motivated by its empirically verified performance: across multiple datasets, it consistently delivers competitive results.
>
> To further assess the effectiveness of the $\ell_{\delta}$-norm, we replaced it in our model with TNN and several representative non-convex surrogates (including Geman[2], Laplace[3], and Schatten $\ell_p$[4]), and carefully tuned the parameters for each alternative. The comparative results are summarized in the table below.
>
> | Datasets | Yale3 | MSRC_v1 | Still-2 | EYaleB10 | COIL20MV | Mfeat | Scene | Scene15 | CIFAR10 |
> |------------|---------|---------------|---------|---------------|-----------------|---------|----------|-------------|-------------|
> | TNN[2] |  20.61±1.29  | 21.62±0.93 | 21.80±1.42 | 15.66±0.40 | 10.36±0.48 | 12.83±0.15 | 14.40±0.19 | 9.82±0.26 | 33.23±1.90
> | Geman[3]| 58.30±2.82 | 96.29±1.09 | 72.55±0.96 | 53.59±1.37 | 67.25±2.25 | 99.20±0.00 | 94.50±0.00 | 55.01±1.16 | 94.92±7.07
> | Lapace[4] | 33.21±3.87 | 41.33±3.06 | 29.42±1.17 | 26.56±1.28 | 46.28±3.99 | 60.51±4.29 | 33.90±0.83 | 18.23±1.16 | 90.13±2.19
> | Schatten$\ell\_p$[5] | 75.76±6.64 | 93.99±1.65 | 87.11±4.95 | 57.09±5.96 | 81.78±12.71 | 98.61±0.13 | 70.43±0.00 | 76.23±4.10 | 33.44±1.19
> | $\ell \_δ$ [1]| 61.82±2.27 | 97.90±0.26 | 74.78±0.46 | 55.72±5.42 | 70.32±1.33 | 96.35±6.62 | 95.42±0.00 | 57.04±3.30 | 93.96± 8.10
>
> The experimental results clearly show that, compared with the traditional TNN, employing any non-convex surrogate within the CAMEL framework leads to improved clustering performance. This demonstrates that non-convex penalties, by adaptively treating singular values, effectively preserve informative components while suppressing noise. Moreover, although the performance of different non-convex functions varies across datasets, the proposed $\ell_{\delta}$-norm consistently exhibits strong competitiveness: it achieves the best performance on MSRC_v1 and Scene, and ranks second on datasets such as Yale3, Still-2, EYaleB10, COIL20MV, Scene15, and CIFAR10. These findings confirm that adopting the $\ell_{\delta}$-norm for tensor low-rank modeling is both reasonable and effective.
> We would also like to clarify that the primary contribution of our work does not lie in designing a particular non-convex surrogate. Rather, the main novelty is our constant-degree–driven connectivity constraint integrated with a low-rank constraint on the latent embedding tensor. The tensor rank surrogate itself is modular within our framework, allowing different functions to be seamlessly incorporated without altering the overall architecture. The experimental evidence demonstrates the extensibility and flexibility of our method: it accommodates various tensor low-rank surrogates while maintaining linear computational complexity, consistently delivering robust and competitive performance.
>
> **Weakness 2: Some details require further explanation. For example, what is the clear definition of the "post-processing-free" methods? It seems that some post-processing-free methods still requires an extra clustering procedure on the learned features.**
>
> **A2**: We sincerely appreciate your thorough comment. The term “post-processing-free” refers to the fact that the features obtained after optimization can be directly used for k-means clustering to obtain results, without requiring additional operations on these features (e.g., SVD on the anchor subspace representation matrix or performing spectral clustering on the similarity matrix) to generate new features for clustering. We will refine this content in the final version of the manuscript.

---

> > ### Author Response · Authors · 2025-11-26
> >
> > **Weakness 3: The ablation study requires a deeper analyses, and current discussion cannot well illustrate the contribution of each component.**
> >
> > **A3**: We thank the reviewer for the valuable comment. We will strengthen the ablation study analysis in the final version to better demonstrate the contribution of each component. The significant performance improvement when combining both components (as shown in Table 4) demonstrates their complementary nature. Specifically, the connectivity constraint (CAMEL-C) serves as a connected component constraint that explicitly enforces the formation of well-connected clusters in the latent space, effectively preventing the fragmentation of clusters and ensuring that data points from the same category form connected subgraphs. Meanwhile, the tensor constraint (CAMEL-T) operates across multiple views by leveraging the high-order correlations among them, effectively aligning the complementary information from different views into a coherent tensor representation. This dual mechanism allows CAMEL-C to guarantee the structural connectivity of clusters within each view's embedding space, while CAMEL-T captures the global consensus and shared patterns across views, creating a synergistic effect where intra-view connectivity and cross-view consistency mutually enhance each other. Together, they provide a more robust and comprehensive representation learning framework that neither component could achieve alone.
> >
> > **Weakness 4: Some typos should be corrected.**
> >
> > **A4**:We thank the reviewer for pointing this out. We will carefully proofread the entire manuscript and correct all typographical errors in the final version.
> >
> > References
> >
> > [1] Z. Kang, C. Peng, and Q. Cheng, “Robust PCA via nonconvex rank approximation,” in Proc. ICDM, 2015, pp. 211–220.
> >
> > [2] Y. Xie, D. Tao, W. Zhang, Y. Liu, L. Zhang, and Y. Qu, “On unifying multi-view self-representations for clustering by tensor multi-rank minimization,” International Journal of Computer Vision, vol. 126, no. 11, pp. 1157–1179, 2018.
> >
> > [3] D. Geman and C. Yang, “Nonlinear image recovery with half-quadratic regularization,” IEEE Transactions on Image Processing, vol. 4, no. 7, pp. 932–946, 1995.
> >
> > [4] Y. Chen, S. Wang, X. Xiao, Y. Liu, Z. Hua, and Y. Zhou, “Self-paced enhanced low-rank tensor kernelized multi-view subspace clustering,” IEEE Transactions on Multimedia, vol. 24, pp. 4054–4066, 2021.
> >
> > [5] W. Xia, Q. Gao, Q. Wang, X. Gao, C. Ding, and D. Tao, “Tensorized bipartite graph learning for multi-view clustering,” IEEE Transactions on Pattern Analysis and Machine Intelligence, vol. 45, pp. 5187–5202.

---

### Meta-Review · Area_Chair_wgbP · 2026-01-13

**Summary:**

This paper receives three positive scores and one negative score. Reviewer bSK1 concerns its shallow ablation study, unclear term explanation, and some typos. Reviewer MEf5 concerns the scale of datasets used in experiments and the model's extendibility.  Reviewer q5uJ concerns its unclear anchor complexity. Also, there are some repeated statements. Reviewer Q7LQ holds that its parameter sensitivity analysis is not clearly justified, and the construction of incomplete multi-view data is insufficiently described.



Reviewer bSK1 acknowledges its detailed theoretical analyses and well-written presentation. Reviewer MEf5  acknowledges its well-organized structure. Reviewer q5uJ acknowledges its extensive experiments. Reviewer Q7LQ  acknowledges its superior clustering performance. The negative Reviewer MEf5 mainly concerns its matrix function, sparse noise adaptability, nonconvex surrogates, and small-scale testing datasets.

**Reviewer Concerns:**

After rebuttal, some term explanations are clarified, deeper ablation analyses are made, and the parameter sensitivity is further explored. The small-scale problem raised by Reviewer  MEf5 seems to still exist.

**Reviewer Scores:**

Reviewer bSK1 may maintain the original score, as the authors provide more detailed explanation and deeper ablation study. Reviewer MEf5 may maintain the original score, as  the small-scale problem has not been well investigated.  Reviewer q5uJ may maintain the original score owing to the  experimental comparisons between $\ell_{\delta}$ norm and conventional tensor nuclear norm. Reviewer Q7LQ may maintain the original score owing to the experimental results on the superiority of H over Z.

---

### Decision · Program_Chairs · 2026-01-26

Accept (Poster)